# GAF-Pano: Zero-Shot Layout-Controlled Panorama Generation via Global Attention Fusion

## Abstract

Achieving both global semantic coherence and precise local layout control in wide-aspect-ratio panorama generation is an unresolved challenge with potential applications. Existing methods that synchronize independent views to generate panoramas often lack semantic coherence and struggle with fine-grained object placement, resulting in contextual artifacts and fragmented objects. We introduce GAF-Pano, a training-free framework for zero-shot layout-controlled panorama generation. GAF-Pano integrates a Global Attention Fusion mechanism into a pre-trained layout-to-image model. Through a Global Context Synchronization, Fusion, and Dispatch workflow, it periodically aggregates latent features from all local views to construct a unified global context, performs multi-level attention computation over this context to achieve true fusion, and then dispatches the enriched global features back to each view, enabling coherent rendering of complex, holistic layouts. Furthermore, we introduce a conditional positional mask to resolve object repetition artifacts that often arise in large specified regions. On a newly constructed yet challenging benchmark for panoramic layout control, GAF-Pano achieves superior performance in both layout fidelity and semantic coherence, faithfully generating complex panoramic scenes.

## 1 Introduction

In recent years, diffusion models Ho et al. (2020); Song et al. (2021); Dhariwal & Nichol (2021) have brought revolutionary breakthroughs to the field of image generation. A significant frontier within this domain is controllable generation, where users can determine the content and precisely control its spatial layout. On fixed-size square images (typically with a 1:1 aspect ratio), pre-trained Layout-to-Image (L2I) models Li et al. (2023); Zheng et al. (2023b); Wang et al. (2024b) have already demonstrated precise adherence to bounding box instructions. However, extending such control to long-form content like panoramas remains a major challenge. For clarity, we scope the term panorama in this paper to mean wide-aspect-ratio images generated through horizontal extension and view stitching. This notion diverges from the conventional 360° spherical panorama and is chosen to align with our methodological focus on controllable generation over extended two-dimensional canvases. This challenge is also reflected at the data level. Unlike fixed-size images backed by large annotated datasets such as COCO Lin et al. (2014), OpenImages Kuznetsova et al. (2020), panoramic datasets with fine-grained layout annotations are scarce. Consequently, training-free, zero-shot methods for controllable panorama generation is an unexplored yet promising direction.

To achieve controllable generation on an expanded canvas, MultiDiffusion Bar-Tal et al. (2023) introduced a pioneering framework. By fusing joint diffusion paths, MultiDiffusion enables coherent panoramic image generation and rudimentary region-based control. However, both panoramic generation and coarse region-based control inherently suffer from object fragmentation issues, where entities may be inconsistently segmented across different spatial regions, and its control capability lies in applying a standard text-to-image model to different regions, a coarse-grained approach where the base model lacks prior knowledge of complex layouts. To overcome these problems, subsequent research has generally progressed along two paths. One class of methods, such as SyncDiffusion Lee et al. (2023), GVCFDiffusion Sun et al. (2025), and PanoFree Liu et al. (2024a), introduces additional supervisory signals at the view level—for instance, using perceptual loss or guided fusion to

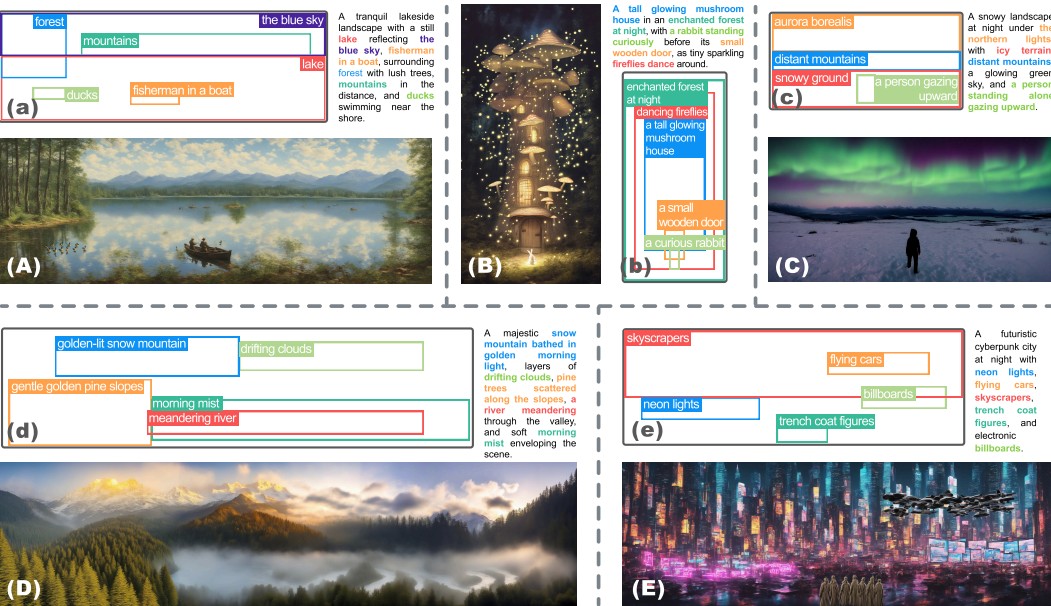

Figure 1: GAF-Pano achieves zero-shot, bounding-box-level, fine-grained layout control on panoramic images with different aspect ratios. The model accurately places objects according to the specified bounding boxes (labeled in lower case), even in complex scenes, resulting in generated images (labeled in upper case).

optimize smooth transitions between views. The other class of methods turns to "in-process intervention" within the model's computational flow. For example, MAD Quattrini et al. (2025) modifies the internal attention structures via "attention fusion"—a mechanism that aggregates Query, Key, and Value matrices into a shared global context to enforce cross-view information sharing. This has been demonstrated to be highly effective in enhancing semantic coherence and has demonstrated stronger modeling capabilities. However, both classes of methods fail to achieve fine-grained layout control while maintaining semantic coherence.

To enhance controllability, we analyzed the underlying mechanism of attention fusion and found that it unlocks a Global Semantic Modeling Capability in pre-trained models. Specifically, it enables the attention mechanism to capture long-range dependencies across panoramic layouts. Visualizations of cross-attention maps (see Figure 3) show that the fused model establishes accurate semantic-to-spatial alignment across the entire canvas.

Based on this insight, we propose GAF-Pano, a framework that applies a Global Attention Fusion mechanism to a pre-trained Layout-to-Image model to achieve fine-grained, globally consistent control. To efficiently implement this within the existing generation paradigm based on fused diffusion paths, we designed a Global Context Synchronization, Fusion, and Dispatch (SFD) workflow that operates directly within the attention layers of the diffusion model. The workflow includes three steps. (1) It periodically synchronizes by aggregating latent features from all overlapping local views into one unified global context. (2) It fuses by performing multi-level attention over this global context to integrate cross-view information. (3) It dispatches by splitting the enriched global context back into each local view path. This process allows us to leverage the model's powerful layout understanding in a zero-shot manner, enabling high-precision control directly within the panoramic generation, as shown in Figure 1. Furthermore, we identified and addressed an emerging issue of object duplication by designing an effective conditional positional mask strategy.

To evaluate our method, we also constructed a new benchmark for layout-controlled generation in panoramas. Results show that our approach surpasses those existing region-based generation methods which are compatible with the MultiDiffusion framework. The main contributions of this paper are summarized as follows:

- Empirically, we analyze the global semantic modeling capability unlocked by the attention fusion mechanism, demonstrating its potential to establish long-range semantic-spatial mappings across an extended canvas.
- We propose GAF-Pano, a framework that integrates the fusion mechanism into pre-trained layout-to-image (L2I) models. To the best of our knowledge, this is the first framework adapting pre-trained L2I models for zero-shot panoramic generation with fine-grained layout control.
- We have constructed a benchmark with a new dataset on layout control for panoramic images. Compared to region-based generation methods in line with MultiDiffusion, our approach achieves advanced performance.

## 2 RELATED WORK

Our research lies at the intersection of layout-controlled synthesis and panorama generation. In the following, we review the representative literature in these two domains and highlight the gaps our work aims to address.

**Layout-Controlled Image Generation.** Layout-controlled image generation aims to synthesize images following specified spatial arrangements and attribute descriptions while preserving visual coherence. Existing diffusion-based methods can be broadly categorized into two groups. *Training-free methods*, such as BoxDiffusion Xie et al. (2023), Layout-Control Chen et al. (2023), and MultiDiffusion Bar-Tal et al. (2023), achieve zero-shot control by guiding attention maps or composing instances via energy functions during inference. *Training-based methods*, including GLIGEN Li et al. (2023), InstanceDiffusion Wang et al. (2024b), MIGC Zhou et al. (2024a), and others Zheng et al. (2023a); Cheng et al. (2024); Wu et al. (2024); Zhou et al. (2024b), introduce learnable modules or specialized architectures (e.g., adapters or separated conditioning branches) to enhance layout adherence. Despite their effectiveness on standard-resolution images, most of these methods are not inherently designed for wide-aspect-ratio inputs. While MultiDiffusion Bar-Tal et al. (2023) has demonstrated initial potential for region-based control on extended canvases, achieving fine-grained layout fidelity and global semantic coherence remains an unresolved challenge, underscoring the need for dedicated panoramic layout generation methods.

**Panorama Generation.** Existing panorama generation methods generally follow two distinct lines: $360°$ *spherical panoramas* and *wide-aspect-ratio 2D panoramas*. Spherical methods, such as PanFusion Zhang et al. (2024), StitchDiffusion Wang et al. (2024a), and DiT360 Feng et al. (2025), utilize specialized architectures to handle equirectangular distortions and ensure rotational continuity, strictly oriented toward immersive full-view scenes. In contrast, wide-aspect-ratio 2D panoramas extend standard diffusion models to long horizontal canvases. Current approaches typically follow either an *iterative extension* paradigm Avrahami et al. (2023); Zhang et al. (2023b); Liu et al. (2024a) or a *joint diffusion* paradigm that fuses overlapping views during sampling (Bar-Tal et al., 2023). Within the joint diffusion framework, MultiDiffusion Bar-Tal et al. (2023) establishes the foundational view-fusion formulation. Subsequent methods have enhanced this pipeline to improve boundary seamlessness: SyncDiffusion Lee et al. (2023) and GVCFDiffusion Sun et al. (2025) introduce perceptual losses and guided/variance-corrected fusion, while SpotDiffusion Frolov et al. (2025) employs temporal window-shifting strategies. MAD Quattrini et al. (2025) incorporated attention fusion to promote cross-view information sharing. However, these techniques primarily target cross-view coherence. When extended to layout-guided generation, they still inherently process each view as an independent crop and lack a mechanism for *global spatial reasoning*. Although spherical panorama models like PanFusion use distance map structural cues, their control focuses on global geometry rather than instance-level placement. Similarly, approaches such as MultiDiffusion with ControlNet Zhang et al. (2023a) or synchronized diffusion frameworks Kim et al. (2024) improve global consistency but do not provide bounding-box–specific grounding. Our method employs global attention fusion within joint diffusion, enabling coherent spatial planning and fine-grained layout control for wide 2D panoramas.

## 3 PRELIMINARIES

**Latent Diffusion Models (LDMs)** Rombach et al. (2022); Podell et al. (2023) (e.g., Stable Diffusion) operate in a latent space obtained via a pretrained VAE $(\mathcal{E}, \mathcal{D})$ Kingma & Welling (2013). A

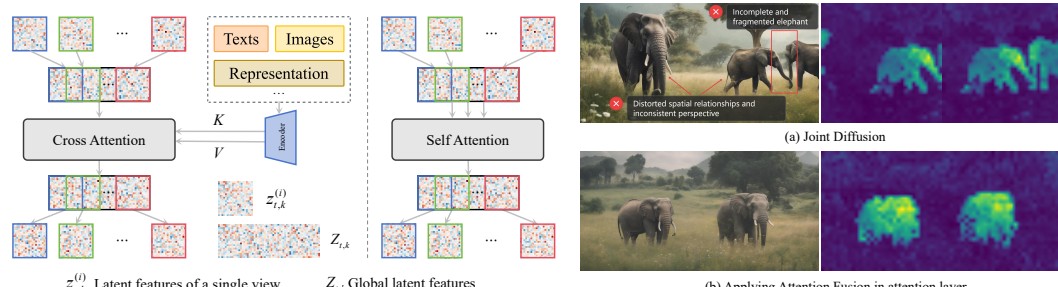

Figure 2: The Attention Fusion Mechanism: by aggregating latent features from multiple views into a global canvas, the model can perform attention on the entire panoramic canvas.

Figure 3: Attention map for the token "elephant". (a) Without attention fusion: view-specific and fragmented. (b) With attention fusion: coherent global representation.

noise prediction network $\epsilon_\theta$ is trained to denoise $z_0 = \mathcal{E}(x_0)$, with objective

$$\mathcal{L} = \mathbb{E}_{t,z_0,\epsilon}\Big[\|\epsilon - \epsilon_\theta(z_t, t, c)\|^2\Big] \tag{1}$$

where $c$ is the conditioning input. To incorporate external conditions like text, LDMs employ cross-attention operation. Layout-controllable models like MIGC Zhou et al. (2024a) and Grounding-Booth Xiong et al. (2024) use **masked cross-attention**:

$$\text{Attention}(Q, K, V, M) = \text{softmax}\left(\frac{QK^\top}{\sqrt{d_k}} + M\right)V, \tag{2}$$

where mask $M$ enforces specific regions to attend only to corresponding textual tokens. Our method, GAF-Pano, extends this mechanism for panoramic generation.

**Joint diffusion.** MultiDiffusion Bar-Tal et al. (2023) generates panoramas by applying a pretrained model to overlapping crops and fusing the outputs at each denoising step through this optimization:

$$\Psi(J_t \mid z) = \arg\min_{J \in \mathcal{J}} \sum_i^n \big\|W_i \odot \big[F_i(J) - \Phi(I_i^t \mid y_i)\big]\big\|^2. \tag{3}$$

Here $W_i$ is a blending mask and $y_i$ the text condition for region $i$.

## 4 PRE-EXPERIMENT

Generating wide-aspect-ratio panoramas faces a fundamental memory limitation. Typical pipelines like MultiDiffusion address this through joint diffusion paths, decomposing panoramas into overlapping square images processed independently, with latent aggregation performed after each denoising step.

We define these independently processed local images as views, with the complete scene formed by spatial integration termed the global canvas. While this view-based approach resolves memory limitations, it introduces a critical challenge, which is to ensure seamless transitions and semantic coherence across overlapped views.

Prior work Quattrini et al. (2025) has shown that fusing attention layers across diffusion paths improves the semantic consistency of panoramic generation, the underlying mechanism remains unexplained. To address this, we conduct a visual and conceptual analysis to uncover how attention fusion enhances consistency, which serves as the empirical motivation for our GAF-Pano framework.

Specifically, we divide the target panoramic canvas $J_T$ into a set of $I$ overlapping local views $\{v_1, v_2, \ldots, v_I\}$, each of standard resolution $h \times w$, where $\bigcup_{i=1}^I v_i = J_T$. At timestep $t$ and layer $k$, the latent features of each view are denoted as $z_{t,k}^{(i)} \in \mathbb{R}^{C \times h \times w}$. Attention fusion then aggregates $\{z_{t,k}^{(i)}\}_{i=1}^I$ into a global latent $Z_{t,k}$ and performs unified attention computation, enabling cross-view semantic alignment and spatial planning at each sampling step (Figure 2).

In standard joint diffusion frameworks (e.g., MultiDiffusion), the self- and cross-attention mechanisms are confined locally within each independent view, preventing cross-view communication. This limitation directly leads to object fragmentation and inconsistent spatial cues. For instance, when generating "a photo of a meadow with an elephant", the attention map for the token "elephant" remains view-specific and fragmented, as visualized in Figure 3(a). This failure to form a holistic plan results in an incoherent final panorama.

In contrast, attention fusion aggregates features from all views into a unified global context, enabling the model to reason over the full canvas. In Figure 3(b), the attention map now accurately localizes the object in the entire panorama.

We refer to this ability as the Global Semantic Modeling Capability, which explains the enhanced consistency and enables us to extend the layout control of pre-trained models to panoramic generation in a zero-shot manner.

## 5 METHOD

### 5.1 PROBLEM DEFINITION

We formalize Layout-Controlled Panorama Generation (LCPG) task as generating an image from a tuple $\mathcal{T}_{LCPG} = (P, \{B, D\})$. This task requires placing objects with both spatial, semantic precision and and stylistic coherence across a wide-aspect-ratio canvas. The task components are:

- $P$: A prompt describing the overall scene or background.
- $B = \{b_1, \ldots, b_N\}$: A set of $N$ bounding boxes specifying the spatial regions of all objects.
- $D = \{d_1, \ldots, d_N\}$: A set of local descriptions, where $d_i$ is the prompt for the corresponding $b_i$.

### 5.2 GAF-PANO FRAMEWORK OVERVIEW

To address the highly challenging task of layout-controlled panorama generation, we have designed the GAF-Pano framework. The core idea of GAF-Pano is to expand the strong capabilities of a pre-trained layout-to-image generation model from local views to the full panoramic scale by employing the proposed Global Attention Fusion mechanism. Instead of training a new model from scratch, we empower an existing layout-to-image model with the ability to perform global planning and precise control on an extended canvas.

This is achieved through a Global Context Synchronization, Fusion, and Dispatch (SFD) workflow that specifically operates within the attention layers of the U-Net during the inference of each sampling step in the whole denoising process. As illustrated in Figure 4, this SFD workflow enables effective global attention fusion across the panoramic canvas by replacing standard independent attention computations.

The SFD workflow operates as follows:

1. **Synchronization (Sync)** is to aggregate the latent features of all views into a unified global latent context.
2. **Fusion (Fuse)** is to apply the designed multi-level attention operations on the global context to achieve information fusion across views.
3. **Dispatch (Dispatch)** is to split the fused global latent back into local views, injecting global context to guide their subsequent generation.

At each diffusion step, we periodically perform three key operations during attention computation as follows. After each denoising step, we apply MultiDiffusion's synchronization step to aggregate the updated latent features, ensuring consistent and coherent panoramic generation.

### 5.3 SYNC-FUSE-DISPATCH WORKFLOW

This process forms the technical core of our method, designed to replace the conventional workflow where U-Net Attention layers process information independently.

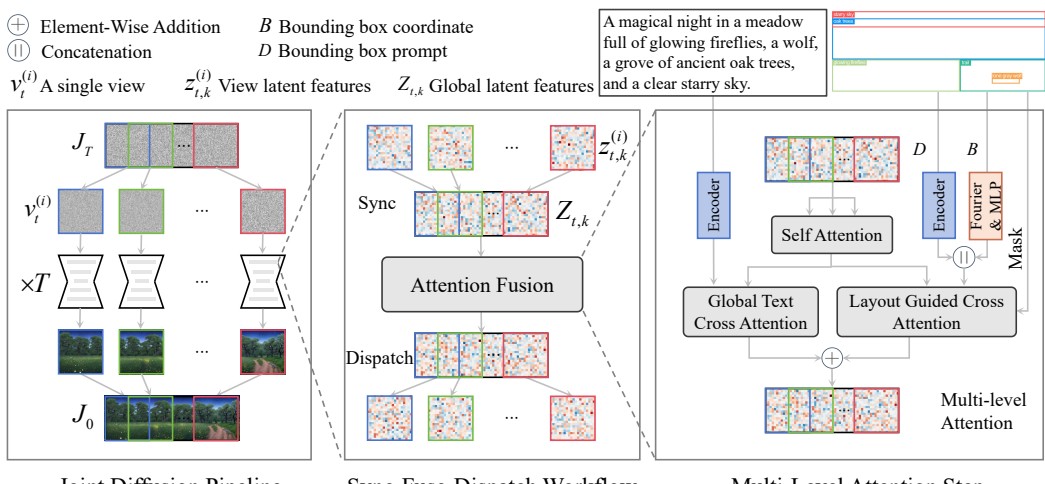

Figure 4: Overview of GAF-Pano. The framework alternates Global Context Synchronization, Multi-Level Attention Fusion, and Context Dispatch to enable globally consistent layout-controlled generation.

### 5.3.1 CROSS-VIEW LATENT SYNCHRONIZATION (SYNC)

The objective of this stage is to aggregate the current state information from all independent views into a single, continuous global context. At a given denoising timestep $t$ and U-Net layer $k$, we first obtain the set of latent features from all $I$ views, $\{z_{t,k}^{(i)}\}_{i=1}^{I}$. We define a Sync operation that maps this set to a global latent $Z_{t,k}$:

$$Z_{t,k} = \text{Sync}(\{z_{t,k}^{(i)}\}_{i=1}^{I}), \quad \forall i \in [1, I] \tag{4}$$

For the latent feature of the overlapping views, we adopt an averaging fusion strategy:

$$Z_{t,k}(p) = \frac{1}{|\mathcal{I}_p|} \sum_{i \in \mathcal{I}_p} z_{t,k}^{(i)}(p) \tag{5}$$

where $\mathcal{I}_p$ denotes the set of indices of all views that contain the spatial position $p$.

### 5.3.2 MULTI-LEVEL ATTENTION IN GLOBAL CONTEXT (FUSE)

The fusion step processes the global latent $Z_{t,k}$ via a multi-level attention block $\Phi_{\text{attn}}$, which inherits its architecture from a pretrained layout-to-image model. It sequentially applies Global Self-Attention (SA) and a Combined Cross-Attention (CA). The update of the global latent representation can be formulated as:

$$Z_{t,k+1} = \Phi_{\text{attn}}(Z_{t,k}, P, (B, D)) \tag{6}$$

The internal attention operations follow the formula:

$$\text{Attention} = \text{Softmax}\left(\frac{QK^{\top}}{\sqrt{d}} + M\right) V \tag{7}$$

Here $Q$, $K$, and $V$ are the query, key, and value matrices, and $M$ is an optional attention mask. The complete attention computation consists of the following two stages.

Stage 1: Global Self-Attention (SA). To capture long-range spatial dependencies and promote structural coherence, SA is applied where the query, key, and value matrices $(Q, K, V)$ are all derived from the global latent $Z_{t,k}$, and no mask is used.

$$Q = ZW_Q, \quad K = ZW_K, \quad V = ZW_V, \quad M = \text{None}. \tag{8}$$

Stage 2: Combined Cross-Attention (CA). Following self-attention, a combination of global-text cross-attention and layout-guided cross-attention is applied to the features.

Global Text Cross Attention (GCA) provides global semantic and stylistic guidance by computing attention between the latent features and the global prompt embedding $E(P)$, and then projected to compute $K$ and $V$. Similarly, no mask is applied:

$$Q = ZW_Q, \quad K = E(P)W_K, V = E(P)W_V, \quad M = \text{None}. \tag{9}$$

Layout Guided Cross Attention (LCA) is aimed for precise layout control. A guidance embedding $G_i$ for each layout region by concatenating the text embedding $E(d_i)$ with a position embedding $E_{\text{pos}}(b_i)$ derived from the bounding box $b_i$ is contructed:

$$G_i = [E(d_i), \text{MLP}(\text{Fourier}(b_i))] \tag{10}$$

The position embedding $E_{\text{pos}}(b_i)$ is computed by encoding the Fourier features of $b_i$ transformed by a multilayer perceptron (MLP). The MLP is already part of the pretrained layout-to-image model Tancik et al. (2020).

This embedding $G_i$ is used to compute $K$ and $V$, while a hard attention mask $M$ derived from $b_i$ enforces spatial constraints.

$$Q = ZW_Q, \quad K = G_iW_K, \quad V = G_iW_V,$$

$$M = \text{Mask}(b_i), \quad \text{Mask}(b_i) = \begin{cases} 0 & \text{if } p \in b_i, \\ -\infty & \text{otherwise} \end{cases} \tag{11}$$

We observe a fundamental dilemma when dealing with large bounding boxes (Figure 5). For boxes designed to depict a single object (such as a cat), uniformly distributed attention often leads to object duplication within the region. In contrast, boxes meant for multiple objects or scenes (such as trees) require uniform attention to ensure coverage and completeness.

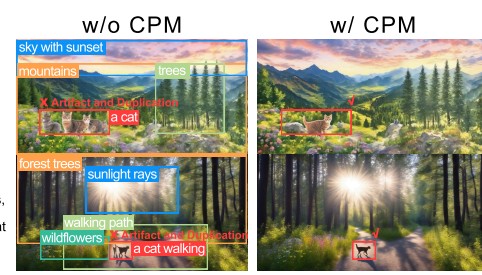

Figure 5: Resolving object duplication with Conditional Position Mask (CPM).

To address this, we propose a Conditional Position Mask (CPM) strategy that modulates the contribution of layout-guided cross-attention based on the semantic content of each bounding box $b_i$. Rather than applying the mask within the attention computation, CPM acts as a spatial weighting factor when combining the outputs of global-text cross-attention and layout-guided cross-attention. Formally, to compute $\text{CPM}_i$ the mask of the $i^{th}$ bounding box, the value $\text{CPM}_i(p)$ at pixel location $p$ is defined as:

$$\text{CPM}_i(p) = \begin{cases} \exp\left(-\frac{u_p^2 + v_p^2}{2\sigma^2}\right) & \text{if } p \in b_i \text{ and } i \in S_{\text{single}}, \\ 1 & \text{if } p \in b_i \text{ and } i \in S_{\text{multi}}, \\ 0 & \text{if } p \notin b_i. \end{cases} \tag{12}$$

Here $(u_p, v_p)$ are normalized coordinates within bounding box $b_i$. $S_{\text{single}}$ and $S_{\text{multi}}$ are sets of indices for single-object and multi-object prompts, respectively. We design an agent to identify those prompts describing multiple objects, which are assigned to $S_{\text{multi}}$, while others default to $S_{\text{single}}$.

The final composite cross-attention computation is then expressed as:

$$\text{Attn}_{\text{cross}} = \text{GCA}_P(Z) + \sum_{i=1}^{N} \text{CPM}_i \odot \text{LCA}_{d_i}(Z) \tag{13}$$

### 5.3.3 GLOBAL CONTEXT DISPATCH (DISPATCH)

After the global attention fusion, the updated global latent $Z_{t,k}$ contains a unified plan for the entire scene. This stage is responsible for dispatching that global plan back to each independent local view path. We define a splitting operation, denoted as $\text{Split}$, which acts as the inverse of the $\text{Sync}$ operation:

$$z_{t,k}^{(i)} = \text{Split}(Z_{t,k}, v_i), \quad \forall i \in [1, I] \tag{14}$$

Each updated local latent $z_{t,k}^{(i)}$ then proceeds through the standard operations of the U-Net's subsequent layers (e.g., convolution, normalization):

$$z_{t,k+1}^{(i)} = \text{UBlock}(z_{t,k}^{(i)}) \tag{15}$$

In summary, our proposed SFD workflow is integrated into MultiDiffusion, where both Global Self-Attention (SA) and Cross-Attention (CA) components are configured to operate during the sampling process.

## 6 EXPERIMENT

### 6.1 EXPERIMENT SETTING

**Benchmark.** To evaluate layout control in panoramic settings, we introduce *Pano-Layout-Bench*, comprising 1,341 unique layout-prompt pairs. The dataset was constructed via a semi-automated pipeline: initial scene descriptions and bounding boxes were generated by a multimodal LLM (GPT-4o) using designed templates, followed by manual refinement to ensure logical coherence and spatial realism. The benchmark covers three aspect ratios: 1:2 (412 samples), 1:3 (456 samples), and 1:4 (473 samples). Regarding layout statistics, the dataset features an average of 4.86 bounding boxes per image, with box counts ranging from 2 to 8 to cover diverse scenarios. The spatial arrangement is also varied; on average, each bounding box occupies approximately 15.4% of the canvas area, with a mean normalized width of 0.44 and height of 0.26. Further details on scene categories and object distributions are provided in Appendix A.9.

**Baselines.** We compare GAF-Pano against several layout-controlled methods, including MultiDiffusion Bar-Tal et al. (2023), SyncDiffusion Lee et al. (2023), and MAD Quattrini et al. (2025). See appendix A.1.2 for how these baseline methods are used for layout control generation.

**Evaluation Metrics.** We evaluate all methods across four critical dimensions:

- **Layout Fidelity:** We compute mIoU, AP, AP50, and AR by comparing generated objects against specified bounding boxes using GroundingDINO Liu et al. (2024b).
- **Text-Image Consistency:** We use CLIP Score Hessel et al. (2021) for global text-image alignment and Local CLIP Score for region-specific consistency with local descriptions.
- **Stylistic Coherence:** We employ Intra-LPIPS Zhang et al. (2018) to measure perceptual similarity across overlapping regions, with lower scores indicating smoother visual transitions.
- **Visual Quality:** We use the LAION Aesthetics Predictor Schuhmann et al. (2022) to assess overall visual appeal and aesthetic quality of the generated panoramas.

**Implementation Details.** All methods are built on the Stable Diffusion XL Podell et al. (2023) backbone. Our GAF-Pano integrates IFAdapter Wu et al. (2024), a layout-to-image model that employs layout-guided masked cross-attention for spatial control within the UNet framework. The $\sigma$ in CPM is set to be 0.15. For the self attention fusion duration, it operates for the first 10 steps, and cross attention operates throughout the entire process. Self-attention fusion and text cross attention fusion are applied at every layer of the UNet, while layout-guided cross attention follows the design of the underlying layout-to-image model and is applied only at the layers specified therein. For all experiments, the denoising process consists of 30 sampling steps. For the baseline methods in line with MultiDiffusion, the boostrapping stage is 10, a parameter intended to allow the generated content to more closely fit the exact bounding box.

### 6.2 RESULTS

**Quantitative Results.** Table 1 demonstrates GAF-Pano outperforms existing methods across most evaluation metrics under the background-only prompt setting. Our method achieves the best layout fidelity across all metrics (mIoU, AP, AP50, AR), representing substantial improvements over baseline methods. For text-image consistency, GAF-Pano attains the highest CLIP and Local CLIP scores, and demonstrates superior stylistic coherence with the lowest Intra-LPIPS. GAF-Pano maintains competitive visual quality while excelling in layout precision and content consistency.

We also report results with holistic prompts (GAF-Pano*). In existing methods, a background bounding box is set to describe area outside the object boxes but on the canvas. This box is equipped

with a prompt with background information only. We propose to use holistic prompts instead, which is a summary of both background and all object descriptions. The holistic setting substantially improves text-image alignment, recall, and visual quality, but leads to a trade-off in precision metrics, likely due to the model generating objects beyond the specified bounding boxes when provided with richer semantic descriptions. Further analysis of these prompt strategies is provided in the appendix A.2.

Table 1: Quantitative comparison of different methods on Pano-Layout-Bench. ↑ indicates higher is better, and ↓ indicates lower is better. The best results are highlighted in **bold**. For fairness, all methods including ours are evaluated with background-only prompts. Holistic prompt settings indicated by gray * are shown for reference.

| Method | Layout Fidelity | | | | Text-Image Consistency | | Stylistic Coherence | Visual Quality |
|--------|-------|-------|---------|-------|--------|-------------|-------------------|----------------|
| | mIoU ↑ | AP ↑ | AP50 ↑ | AR ↑ | CLIP ↑ | Local CLIP ↑ | Intra-LPIPS ↓ | Aesthetic Score ↑ |
| MultiDiffusion | 0.57 | 0.17 | 0.29 | 0.37 | 30.07 | 25.91 | 0.6865 | 5.89 |
| SyncDiffusion | 0.52 | 0.13 | 0.25 | 0.31 | 29.10 | 25.06 | 0.6625 | **6.07** |
| MAD | 0.57 | 0.21 | 0.35 | 0.38 | 29.44 | 25.96 | 0.6007 | 5.79 |
| **GAF-Pano** | **0.63** | **0.25** | **0.44** | **0.44** | **30.59** | **26.74** | **0.5665** | 5.81 |
| GAF-Pano* | 0.70 | 0.25 | 0.44 | 0.49 | 32.37 | 27.45 | 0.6038 | 6.12 |

**Qualitative Results.** Figure 6 showcases the qualitative results of GAF-Pano compared to the baseline methods. We can see that GAF-Pano generates panoramas with precise object placements according to the specified bounding boxes, while maintaining semantic coherence and stylistic consistency across the entire scene. In contrast, the baseline methods often struggle with object misalignment, inconsistent details, and fragmented contexts.

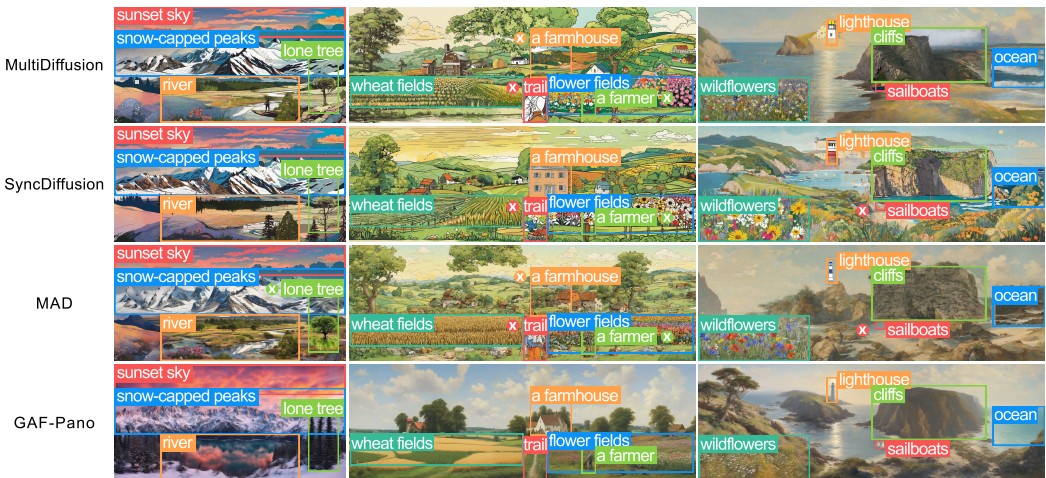

Figure 6: Qualitative comparison of generated panoramas with baseline methods using background prompt. GAF-Pano aligns objects accurately with the layout and preserves global consistency, whereas baselines suffer from noticeable misalignment and visual inconsistency. Best viewed magnified on screen.

## 6.3 ABLATION STUDY

We ablate the Global Self-Attention (SA) fusion by varying its duration in the first denoising steps ($t = 0, 10, 20, 30$), while keeping the combined Cross-Attention (CA) active throughout for consistent semantic and layout guidance. As shown in Table 2, disabling SA ($t = 0$) greatly reduces stylistic coherence (highest Intra-LPIPS), confirming its role in global coherence modeling. Longer SA usage improves coherence (lower Intra-LPIPS) and layout accuracy (higher AP/AP50), but slightly decreases text-image alignment (CLIP Score). We adopt $t = 10$ as it achieves the best trade-off, establishing macro-structure early while preserving overall performance. Additional ablations are provided in Appendix A.6.

Table 2: Ablation results of applying Global Self-Attention (SA) fusion for different durations $t$ during denoising.

| $t$ (steps) | Layout Fidelity | | | | Text-Image Consistency | | Stylistic Coherence | Visual Quality |
|---|---|---|---|---|---|---|---|---|
| | mIoU ↑ | AP ↑ | AP50 ↑ | AR ↑ | CLIP ↑ | Local CLIP ↑ | Intra-LPIPS ↓ | Aesthetic Score ↑ |
| $t = 0$ | **0.68** | 0.21 | 0.37 | 0.45 | **32.34** | **27.71** | 0.6140 | **6.18** |
| $t = 10$ | **0.68** | 0.24 | 0.42 | 0.45 | 32.19 | 27.62 | 0.5752 | 6.12 |
| $t = 20$ | **0.68** | 0.24 | 0.43 | **0.46** | 31.83 | 27.55 | 0.5613 | 6.07 |
| $t = 30$ | 0.67 | **0.26** | **0.46** | 0.43 | 31.53 | 27.36 | **0.5478** | 6.15 |

## 7 CONCLUSION

In this paper, we propose GAF-Pano, a training-free, zero-shot framework designed to address the dual challenges of precise layout control and global semantic coherence in wide-aspect-ratio panorama generation. Through a novel Global Context Synchronization, Fusion, and Dispatch work-flow, we integrate a Global Attention Fusion mechanism into a pre-trained layout-to-image model. This mechanism enables global semantic modeling on an extended canvas by performing unified attention computation within a global context, thereby achieving holistic planning and fine-grained control over complex scenes. For rigorous evaluation, we constructed the Pano-Layout-Bench benchmark. Experimental results demonstrate that GAF-Pano significantly outperforms existing methods in layout fidelity, text-image consistency, and visual coherence. Overall, our framework offers a practical and effective approach to controllable long-form image generation.

## 8 ETHICS STATEMENT

Our method inherits potential biases from pre-trained layout-to-image models, which may reflect social or cultural stereotypes. It can also be misused to generate fake or misleading images. Furthermore, the algorithm might rely on artworks from human painters without proper authorization, raising concerns about intellectual property and consent.

In addition, the benchmark used in our experiments is partially generated by large language models (LLMs), which may contain biases or inaccuracies. We caution against uncritical use of such data.

For our user study, we ensured that all participants provided informed consent, and their responses were anonymized to protect privacy. No participants were exposed to harmful content during the study.

## 9 REPRODUCIBILITY STATEMENT

We have taken several steps to ensure the reproducibility of our work. Appendix A.1 provides a thorough discussion justifying our choice of the Joint Diffusion framework and examines the implications and practical considerations of directly applying pre-trained Layout-to-Image models for panorama generation. We also describe how baseline methods control generation. Additional ablation studies and quantitative comparisons are presented in Appendix A.6 A.10. The construction of our benchmark dataset, together with detailed statistics is provided in Appendix A.9. Finally, We also provide a discussion of the limitations of our method and potential directions for improvement (Appendix A.11). We will make our code publicly available on GitHub to facilitate reproducibility and further research.

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

# A APPENDIX

## A.1 DISCUSSION

### A.1.1 JUSTIFICATION FOR THE JOINT DIFFUSION FRAMEWORK

We justify our joint diffusion framework by contrasting it with direct generation approaches, specifically focusing on the integration of the IFAdapter Wu et al. (2024). Attempting to generate a wide-aspect-ratio panorama in a single pass presents a significant Out-of-Distribution (OOD) task for Layout-to-Image (L2I) models pre-trained on square resolutions. This training-inference mismatch primarily leads to Layout Collapse, where the model's spatial reasoning fails over the extended canvas, resulting in inaccurate object placement and structural incoherence.

To address this, GAF-Pano employs a Joint Diffusion strategy. Unlike direct inference which stretches the model's capacity, our approach constructs a unified global context by synchronizing information across local views. This resolves the issue of information fragmentation—where large bounding boxes are split across views—empowering the model to execute fine-grained layout control globally.

**Combined Analysis of Quality and Fidelity.** The superiority of this approach is evidenced by the joint analysis of the quantitative results in Table 3 and qualitative comparisons in Figure 7. While the direct application of the IFAdapter acts as a strong baseline, it still struggles with the domain gap on panoramic canvases. As shown in the visual comparison, GAF-Pano significantly enhances generation quality by producing richer local details and fewer visual artifacts. This qualitative observation aligns with our quantitative metrics: GAF-Pano outperforms the direct IFAdapter in Layout Fidelity (mIoU $0.69 \rightarrow \mathbf{0.70}$) and achieves a higher Aesthetic Score ($5.97 \rightarrow \mathbf{6.12}$).

These improvements indicate that GAF-Pano does not merely enforce layout constraints but actively improves the generative quality. By processing local views within the model's native resolution and fusing them coherently, it offers both precise control and superior visual fidelity.

Table 3: Quantitative comparison between Direct Generation and GAF-Pano. We evaluate three baseline models generating panoramas directly versus our GAF-Pano framework. GAF-Pano demonstrates superior performance in layout fidelity and visual quality (Aesthetic Score), confirming that our joint diffusion strategy effectively reduces artifacts and improves details compared to direct inference.

| Method | Layout Fidelity | | | | Text-Image Consistency | | Stylistic Coherence | Visual Quality |
|---|---|---|---|---|---|---|---|---|
| | mIoU ↑ | AP ↑ | AP50 ↑ | AR ↑ | CLIP ↑ | Local CLIP ↑ | Intra-LPIPS ↓ | Aesthetic Score ↑ |
| InstanceDiffusion | 0.46 | 0.11 | 0.16 | 0.25 | 30.56 | 24.49 | 0.6031 | 5.46 |
| IFAdapter | 0.69 | **0.26** | **0.47** | 0.48 | 31.53 | 27.20 | **0.5749** | 5.97 |
| CreatiLayout | 0.34 | 0.03 | 0.06 | 0.12 | 32.32 | 23.67 | 0.5951 | 5.73 |
| **GAF-Pano (Ours)** | **0.70** | 0.25 | 0.44 | **0.49** | **32.37** | **27.45** | 0.6038 | **6.12** |

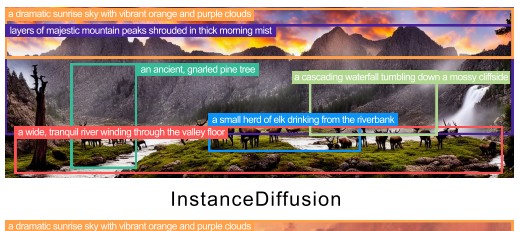

InstanceDiffusion

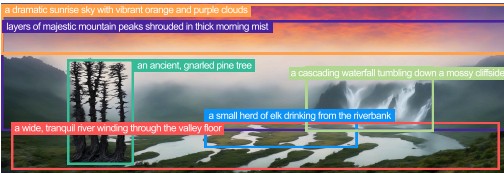

IFAdapter

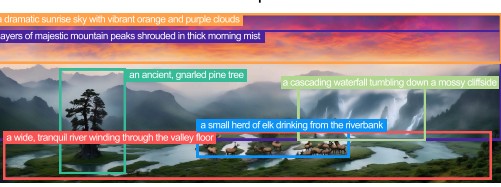

CreatiLayout

GAF-Pano

Figure 7: Qualitative comparison of Direct Generation vs. GAF-Pano. Compared to direct generation methods (InstanceDiffusion, IFAdapter, CreatiLayout), GAF-Pano not only maintains better layout fidelity but also generates images with more details and fewer artifacts. It also ensures that objects are rendered completely and coherently across the 1:3 panoramic aspect ratio.

### A.1.2 COMPARISON WITH MULTIDIFFUSION (MD) REGION GENERATION

MultiDiffusion Bar-Tal et al. (2023) proposes the Follow-the-Diffusion-Paths (FTD) optimization problem, which achieves consistent fusion of multiple diffusion paths by minimizing the following loss function:

$$\mathcal{L}_{FTD}(J|J_t, z) = \sum_{i=1}^{n} \|W_i \odot [F_i(J) - \Phi(F_i(J_t)|y_i)]\|^2 \tag{16}$$

where $J_t$ is the target image at time step $t$, $F_i(J_t)$ is the image space mapping function, $\Phi$ is the pre-trained diffusion model, $W_i$ is the pixel weight matrix.

The analytical solution to this optimization problem is:

$$J_{t-1} = \frac{\sum_{i=1}^{n} F_i^{-1}(W_i) \odot F_i^{-1}(\Phi(F_i(J_t)|y_i))}{\sum_{i=1}^{n} F_i^{-1}(W_i)} \tag{17}$$

In implementation, the panoramic image is divided into $I$ overlapping view windows $\{V_1, V_2, \ldots, V_I\}$ in the latent space, including two practical applications:

**Panorama Generation.** Generate high-resolution panoramic images $J \in \mathbb{R}^{H' \times W' \times C}$ from a single text prompt $y$, where $H' \gg H, W' \gg W$. Each view $V_i$ shares the text prompt $y$ and performs independent denoising:

$$\hat{V}_i = \Phi(V_i|y, t) \tag{18}$$

The final value of each pixel index $p$ is obtained through weighted averaging of all view results covering that position to get $J_{t-1}$:

$$J_{t-1}(p) = \frac{1}{|\mathcal{I}(p)|} \sum_{i \in \mathcal{I}(p)} \hat{V}_i(p) \tag{19}$$

where $\mathcal{I}(p)$ denotes the set of views covering pixel $p$, or gradient weights can be used for fusion.

**Region-Controllable Image Generation.** Given a set of region masks $\{M_k\}_{k=0}^m \subset \{0,1\}^{H \times W}$ and corresponding text conditions $\{y_k\}_{k=0}^m$, generate images satisfying spatial semantic constraints.

Similarly, the extended canvas is divided into overlapping views $V_i$. For the $i^{th}$ view, local masks $M_{i,k}$ are defined. Within each view $V_i$, the view is replicated $m$ times and denoised in parallel for all semantic conditions $\{y_k\}_{k=0}^m$:

$$\{\hat{V}_i^{(k)}\}_{k=0}^m = \{\Phi(V_i^{(k)}|y_k, t)\}_{k=0}^m \tag{20}$$

Then, fusion is performed based on all generation results within views and their corresponding masks to obtain the final image:

$$J_{t-1}(p) = \frac{\sum_{i \in \mathcal{I}(p)} \sum_{k=0}^m M_{i,k}(p) \odot \hat{V}_i^{(k)}(p)}{\sum_{i \in \mathcal{I}(p)} \sum_{k=0}^m M_{i,k}(p)} \tag{21}$$

**The Implementation of baselines.** The baselines of MD, SyncDiffusion Lee et al. (2023), and MAD Quattrini et al. (2025) adopt region-controllable generation pipelines that independently synthesize the content inside each bounding box and subsequently fuse the results. SyncDiffusion and MAD function as plug-and-play modules within this region-controlled setting, enhancing coherence without providing more explicit spatial control.

In contrast, our method builds upon a panoramic generation paradigm and integrates layout cross-attention from pre-trained layout-to-image models into our Sync–Fuse–Dispatch workflow, enabling global coordinated planning and semantically consistent layout control across the extended canvas.

From the perspective of fusion hierarchy and mechanisms, existing MultiDiffusion-based methods primarily operate in the sample space: (1) they perform direct fusion of intermediate noisy samples at each denoising step, constituting low-level fusion at the sample level; (2) the fusion process only considers local spatial constraints without semantic understanding of text prompts, leading to common issues such as boundary artifacts and visual inconsistencies. In contrast, our method performs fusion in the hidden feature space of the denoising network, achieving feature integration at a higher abstraction level. Through self-attention mechanisms, we realize global view consistency fusion while utilizing cross-attention mechanisms to fully consider semantic constraints from text prompts, demonstrating superior performance in maintaining global layout accuracy and structural consistency.

## A.2 EFFECT OF PROMPT TYPES

In our Layout-to-Image Generation setting, each sample is defined as a tuple $\mathcal{T}_{\mathrm{LCPG}} = (P, \{B, D\})$, where $P$ is a global prompt describing the scene, $B = \{b_1, \ldots, b_N\}$ is a set of bounding boxes specifying object locations, and $D = \{d_1, \ldots, d_N\}$ is a corresponding set of object prompts. Although both prompt types use the same layout information $\{B, D\}$ to control spatial placement and local semantics, they differ in how the global prompt $P$ is formulated and integrated with the layout.

Although both prompt types use the same layout information $\{B, D\}$ to control spatial placement and local semantics, they differ in how the global prompt $P$ is formulated and integrated with the layout.

**Background-Only Prompt Setting.** In the background-only approach (as exemplified by MultiDiffusion), the prompt structure consists of:

- Background Prompt ($P_{bg}$) : A descriptive prompt focusing solely on the scene background or environmental context. (e.g., "A quiet forest scene.")

- Box-level prompts ($D$): Provide individual object semantics associated with each bounding box. (e.g., "a wooden cabin", "a dirt path")
- Spatial Constraints ($B$): Bounding boxes defining object placement.

**Holistic Prompt Setting.** The holistic approach integrates all scene elements into a unified prompt structure:

- Holistic Prompt ($P_h$): A complete description encompassing both background and foreground elements in their intended context. (e.g., "A quiet forest scene with a cabin and a dirt path.")
- Box-level prompts ($D$): Provide individual object semantics associated with each bounding box. (e.g., "a wooden cabin", "a dirt path")
- Spatial Constraints ($B$): Bounding boxes defining object placement.

Figure 8: Comparison of background and holistic prompt settings. MultiDiffusion shows fragmented and duplicated objects under holistic prompts due to prompt leakage, while our method maintains spatial consistency and reduces out-of-box generation.

As shown in Figure 8, when using the Holistic Prompt Setting, MultiDiffusion exhibits notable limitations, including fragmented object structures and object duplication, and generation of visual artifacts outside the specified bounding box regions. These issues arise due to holistic prompt leakage, where globally described objects (e.g., "a wooden cabin") are redundantly instantiated across multiple views, even if only a single bounding box is provided. The region-based fusion mechanism in MultiDiffusion lacks explicit global coordination, leading to spatial inconsistencies, structural collisions, and visually implausible compositions when handling comprehensive scene descriptions. In contrast, our proposed method integrates the holistic prompt more effectively by maintaining semantic coherence across the entire canvas and constraining object generation within intended boundaries. While occasional out-of-box generation may still occur, our approach significantly reduces fragmentation artifacts and demonstrates superior spatial-semantic alignment, preserving both visual quality and layout fidelity.

## A.3 Efficiency and Scalability Analysis

We conducted a comprehensive analysis of both computational efficiency and generation stability across varying numbers of bounding boxes ($N = 2, 6, 10$). Experiments were conducted on an NVIDIA vGPU (48GB) generating 1:2 aspect ratio panoramas.

**Computational Scalability ($O(1)$ vs. $O(N)$).** As illustrated in Table 4, existing region-based methods suffer from severe computational bottlenecks as layout complexity increases. MultiDiffusion, while efficient for sparse layouts (59s for $N = 2$), exhibits a drastic slowdown for dense layouts, increasing by 281% to 225s for $N = 10$. Similarly, MAD and SyncDiffusion show linear scaling with the number of bounding boxes, reaching 250s and 277s respectively.

In sharp contrast, GAF-Pano demonstrates remarkably stable inference speeds, operating in effectively constant time ($O(1)$) with respect to layout complexity. Our inference time increases

marginally from 63.6s ($N = 2$) to 64.8s ($N = 10$)—a negligible rise of only 1.9%. This efficiency stems from our architecture: unlike baseline methods that require generating objects separately, GAF-Pano processes all layout conditions in parallel via the layout-guided cross-attention mechanism within a single global context fusion pass.

Table 4: Comparison of Inference Time (seconds) with Varying Layout Complexity. While baselines slow down significantly as the number of bounding boxes ($N$) increases, GAF-Pano maintains near-constant inference speed.

| Method | $N = 2$ | $N = 6$ | $N = 10$ | Relative Increase (%, $N = 2 \rightarrow 10$) |
|---|---|---|---|---|
| MultiDiffusion | **59.2** | 142.5 | 225.5 | +281.0% |
| MAD | 70.9 | 160.4 | 249.7 | +252.1% |
| SyncDiffusion | 111.6 | 193.8 | 277.6 | +148.7% |
| **GAF-Pano (Ours)** | 63.6 | **63.8** | **64.8** | **+1.9%** |

**Memory Overhead Profiling.** To pinpoint the resource bottlenecks within our Sync-Fuse-Dispatch (SFD) workflow, we conducted a detailed layer-wise memory profiling, quantifying the "Peak Memory Overhead"—defined as the maximum transient memory increase observed during a specific stage relative to the pre-stage state. As visualized in Figure 9, the memory consumption patterns reveal a counter-intuitive insight regarding the cost of global fusion.

Contrary to the assumption that global attention computation is the primary resource sink, our profiling identifies the Synchronization (Sync) stage as the actual memory bottleneck. At high-resolution layers (e.g., Down/Up Block 1), Sync operations incur a peak overhead of approximately 660 MB. This is attributed to the "double buffering" required to retain multi-view feature tensors while simultaneously allocating the aggregated global canvas.

In contrast, the Fusion (Fuse) stage exhibits remarkably high efficiency, with a negligible overhead of only 40MB to 80MB across layers. This efficiency is achieved by leveraging optimized kernel fusion (e.g., Flash Attention), which prevents the full materialization of the $N \times N$ attention score matrix in HBM. Consequently, the memory complexity of our fusion mechanism is reduced to near-linear $O(N)$, confirming that the primary cost of GAF-Pano stems from the linear buffer allocation in the Sync stage rather than quadratic attention computation.

**Performance Stability (Success Rate)**. Beyond efficiency, we evaluated the stability of control precision using Success Rate @ 50 (SR@50), defined as the percentage of generated objects achieving an Intersection-over-Union (IoU) $> 0.5$ with the ground truth box. As shown in Figure 10, GAF-Pano maintains high robustness. Starting from a high precision of 0.93 for simple scenes, the performance stabilizes around 0.83-0.86 even as the number of phrases increases to 8 or more. The Average IoU similarly remains steady ($\sim 0.71$). This confirms that GAF-Pano does not suffer from performance degradation in dense layouts, successfully balancing high-speed inference with robust layout fidelity.

### A.4 GENERALIZATION TO MASKED CROSS-ATTENTION MECHANISMS

To demonstrate the universality of GAF-Pano, we evaluate its compatibility with the broader class of methods that utilize **layout-guided masked cross-attention** for spatial control. Beyond the SDXL-based IFAdapter used in our main experiments, we integrated GAF-Pano with MIGC Zhou et al. (2024a), a representative method built on Stable Diffusion 1.5.

Like IFAdapter, MIGC relies on injecting spatial constraints into the cross-attention maps. By incorporating our *Sync-Fuse-Dispatch* workflow, we verify that GAF-Pano can successfully extend this attention-masking paradigm from a local single-view context to a global panoramic context, regardless of the underlying backbone architecture.

**Quantitative Analysis.** Table 5 presents the performance of GAF-Pano integrated with MIGC.

- **Mechanism Validity:** The model achieves an mIoU of 0.63 and AP50 of 0.49. These robust layout fidelity scores confirm that our framework effectively synchronizes attention masks across views. It demonstrates that the core principle of masked cross-attention can be seamlessly scaled to panoramic generation via our joint diffusion strategy.

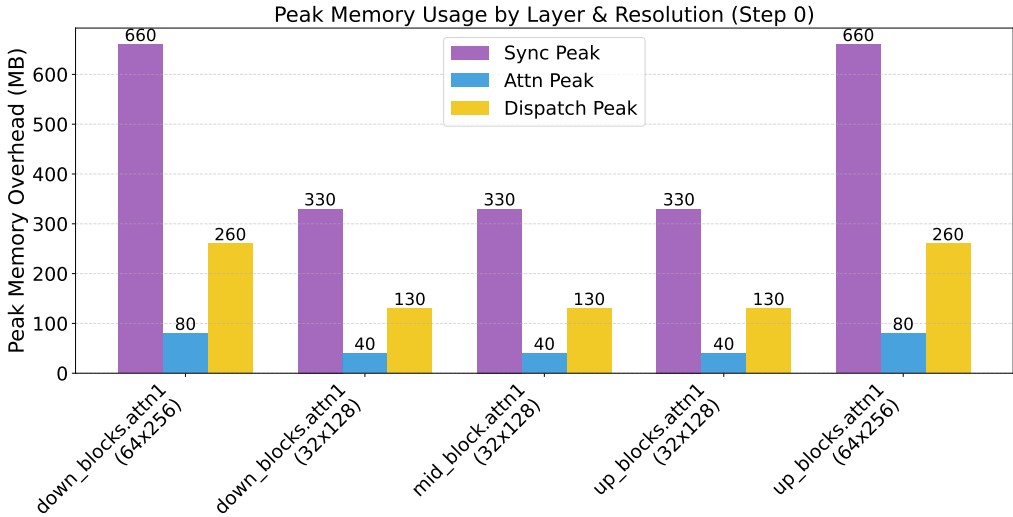

Figure 9: Memory Overhead Analysis of the Sync-Fuse-Dispatch Workflow. We profile the peak transient memory cost across different U-Net layers. Surprisingly, the **Sync** stage (orange) dominates memory usage due to tensor aggregation buffering (∼660MB at peak), while the **Fuse** stage (blue) remains highly efficient (∼80MB) thanks to optimized attention kernels. This demonstrates that Global Attention Fusion does not introduce significant memory penalties.

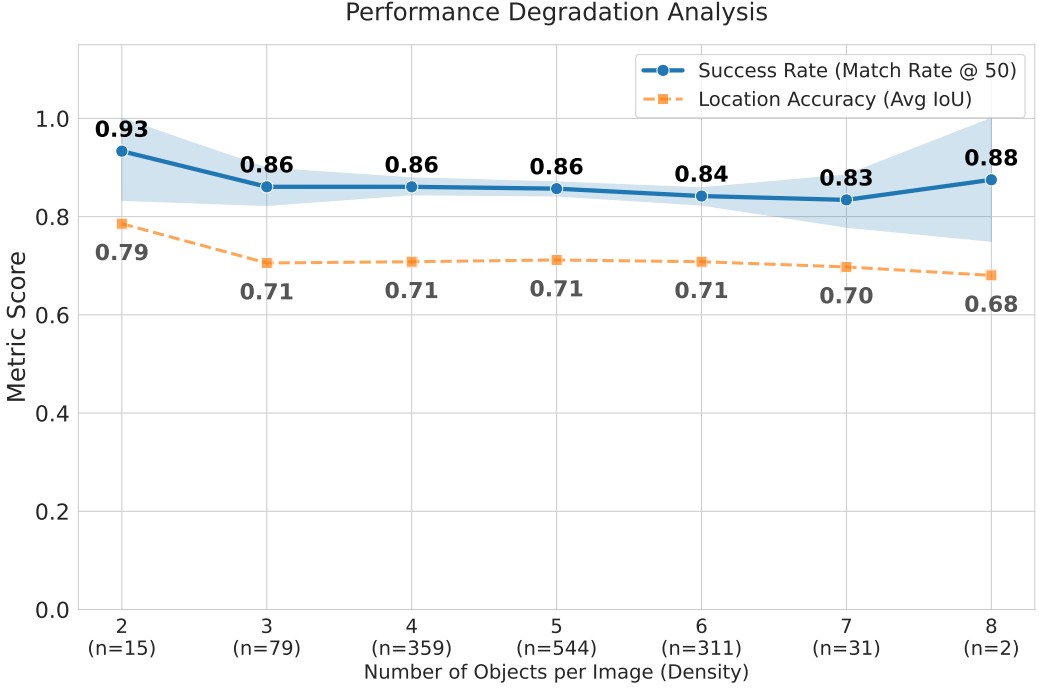

Figure 10: Performance stability of GAF-Pano under increasing scene complexity. Success Rate @ 50 (SR@50) and Average IoU remain consistently high as the number of input phrases grows, demonstrating robustness in dense layouts without significant degradation.

- **Global Coherence:** Despite the change in backbone (to SD1.5), the method maintains high stylistic coherence (Intra-LPIPS 0.5133), proving that the global attention fusion remains stable across different implementations of the attention mechanism.

Table 5: **Quantitative performance of GAF-Pano integrated with MIGC.** The results confirm that our framework is compatible with different models sharing the layout-guided masked cross-attention paradigm.

| Method | Layout Fidelity | | | | Text-Image Consistency | | Stylistic Coherence | Visual Quality |
|---|---|---|---|---|---|---|---|---|
| | mIoU ↑ | AP ↑ | AP50 ↑ | AR ↑ | CLIP ↑ | Local CLIP ↑ | Intra-LPIPS ↓ | Aesthetic Score ↑ |
| GAF-Pano (MIGC) | 0.63 | 0.31 | 0.49 | 0.49 | 29.55 | 25.17 | 0.5133 | 5.45 |

## A.5 PSEUDOCODE FOR THE SFD WORKFLOW

Algorithm 1 shows the pseudocode for our Sync-Fuse-Dispatch Workflow.

---

**Algorithm 1:** The SFD workflow of a single U-Net Layer

---

**Input:** $\{z_{t,k}^{(i)}\}_{i=1}^{I}$;      // Latent features of $I$ views at timestep $t$, start of layer $k$

**Data:** $\mathcal{T}_{LCPG} = (P, \{B, D\})$

**Output:** $\{z_{t,k+1}^{(i)}\}_{i=1}^{I}$;         // features after attention fusion

**Function** Sync($\{z_{t,k}^{(i)}\}$):
> $Z_{t,k}(p) \leftarrow \frac{1}{|I_p|} \sum_{i \in I_p} z_{t,k}^{(i)}(p)$;    // Average overlapping regions (Eq. 5)
> **return** $Z_{t,k}$;    // Global latent tensor for layer $k$

**Function** Fuse($Z_{t,k}, P, \{B, D\}$):
> $Z_{t,k}^{\text{SA}} \leftarrow \text{SA}(Z_{t,k})$;    // Global self attention (Eq. 8)
> $Z_{t,k}^{\text{GCA}} \leftarrow \text{GCA}(Z_{t,k}^{\text{SA}}, E(P))$;    // Global text cross attention (Eq. 9)
> $Z^{\text{LCA-sum}} \leftarrow 0$;
> **for** $i = 1, \ldots, N$ **do**
> > $G_i \leftarrow [E(d_i), \text{MLP}(\text{Fourier}(b_i))]$;    // Layout-guided embedding (Eq. 10)
> > $Z^{\text{LCA-sum}} \leftarrow Z^{\text{LCA-sum}} + \text{CPM}_i \cdot \text{LCA}(Z_{t,k}^{\text{SA}}, G_i, \text{Mask}(b_i))$;    // Masked layout cross attention (Eq. 14)
> $Z'_{t,k} \leftarrow Z_{t,k}^{\text{GCA}} + Z^{\text{LCA-sum}}$;    // Fused global context for layer $k$
> **return** $Z'_{t,k}$

**Function** Dispatch($Z'_{t,k}$):
> **for** $i = 1, \ldots, I$ **do**
> > $z_{t,k}'^{(i)} \leftarrow Z'_{t,k}[v_i]$;    // Crop global to local (Eq. 15)
> **return** $\{z_{t,k}'^{(i)}\}_{i=1}^{I}$

**Function** SFD_Workflow($\{z_{t,k}^{(i)}\}$):
> $Z_{t,k} \leftarrow$ Sync($\{z_{t,k}^{(i)}\}$);
> $Z'_{t,k} \leftarrow$ Fuse($Z_{t,k}, P, \{B, D\}$);
> $\{z_{t,k}'^{(i)}\} \leftarrow$ Dispatch($Z'_{t,k}$);
> $\{z_{t,k+1}^{(i)}\} \leftarrow$ Ublock($\{z_{t,k}'^{(i)}\}$);  // Passed to the remaining U-Net block (UBlock)
> **return** $\{z_{t,k+1}^{(i)}\}$;

---

## A.6 MORE ABLATION STUDIES

### A.6.1 ABLATION ON GLOBAL TEXT CROSS ATTENTION.

We ablate the Global Text Cross Attention (GCA) by varying its application duration across the first denoising steps ($t = 0, 10, 20, 30$). During this study, Self-Attention (SA) fusion is fixed at $t = 10$, and Layout Cross Attention (LCA) is applied throughout. As shown in Table 6, disabling GCA ($t = 0$) results in slightly higher Intra-LPIPS and marginally lower AP50, indicating a small drop in stylistic coherence and layout precision. Overall performance remains stable across different GCA durations. This suggests GCA contributes to global semantic alignment, but its effect is less pronounced than SA or LCA.

Table 6: Ablation results of applying Global Text Cross Attention (GCA) fusion for different durations $t$ during denoising.

| $t$ (steps) | Layout Fidelity | | | | Text-Image Consistency | | Stylistic Coherence | Visual Quality |
|---|---|---|---|---|---|---|---|---|
| | mIoU ↑ | AP ↑ | AP50 ↑ | AR ↑ | CLIP ↑ | Local CLIP ↑ | Intra-LPIPS ↓ | Aesthetic Score ↑ |
| $t = 0$ | 0.66 | **0.22** | 0.38 | 0.43 | 32.05 | 27.59 | 0.5778 | 6.11 |
| $t = 10$ | 0.66 | **0.22** | 0.38 | 0.43 | 32.06 | 27.63 | 0.5775 | 6.11 |
| $t = 20$ | 0.66 | **0.22** | **0.39** | 0.43 | 32.07 | 27.62 | **0.5774** | 6.11 |
| $t = 30$ | 0.66 | 0.21 | **0.39** | 0.44 | **32.10** | **27.64** | **0.5774** | 6.11 |

### A.6.2    ABLATION ON LAYOUT GUIDED CROSS ATTENTION.

We ablate the Layout Guided Cross Attention (LCA) fusion by varying its application duration during the early denoising steps $t = 0, 10, 20, 30$, where LCA is only applied before step $t$. Global Self-Attention (SA) fusion is fixed at $t = 10$, and Global Text Cross Attention is retained throughout the process.

As shown in Table 7, completely disabling LCA fusion ($t = 0$) leads to poor layout fidelity, indicating that spatial guidance is critical for aligning the output with the desired layout. Increasing the duration of LCA fusion progressively enhances layout fidelity and stylistic coherence (lower Intra-LPIPS), while causing a slight decrease in text-image consistency (CLIP Score).

These results suggest that layout-guided cross attention is particularly effective during the early-to-mid stages of denoising, where spatial structure is being established. Longer fusion duration provides better control over layout and style, but must be balanced against potential semantic drift.

Table 7: Ablation results of applying Layout Guided Cross Attention (LCA) fusion for different durations $t$ during denoising.

| $t$ (steps) | Layout Fidelity | | | | Text-Image Consistency | | Stylistic Coherence | Visual Quality |
|---|---|---|---|---|---|---|---|---|
| | mIoU ↑ | AP ↑ | AP50 ↑ | AR ↑ | CLIP ↑ | Local CLIP ↑ | Intra-LPIPS ↓ | Aesthetic Score ↑ |
| $t = 0$ | 0.36 | 0.04 | 0.08 | 0.14 | **33.05** | 24.66 | 0.6222 | **6.60** |
| $t = 10$ | 0.53 | 0.13 | 0.24 | 0.30 | 32.70 | 26.22 | 0.6003 | 6.30 |
| $t = 20$ | 0.63 | 0.20 | 0.35 | 0.39 | 32.40 | 27.24 | 0.5903 | 6.17 |
| $t = 30$ | **0.65** | **0.21** | **0.38** | **0.42** | 32.22 | **27.65** | **0.5806** | 6.10 |

### A.6.3    ABLATION ON CONDITIONAL POSITION MASK.

We further ablate the Conditional Position Mask (CPM) by varying the weighting factor $w$ in $\sigma = w \cdot \min(\text{box\_height}, \text{box\_width})$. As illustrated in Figure 5 and confirmed by the quantitative results in Table 8, introducing CPM reduces object duplication and visual artifacts compared to the baseline (w/o CPM), while maintaining competitive layout fidelity and text-image consistency. A smaller $\sigma$ enforces stronger suppression within single-object regions, improving stylistic coherence and visual quality. As $\sigma$ increases (larger $w$), CPM gradually weakens and the behavior approaches that of the baseline without CPM.

Table 8: Ablation results of Conditional Position Mask (CPM) with different $w$ values.

| $w$ | Layout Fidelity | | | | Text-Image Consistency | | Stylistic Coherence | Visual Quality |
|---|---|---|---|---|---|---|---|---|
| | mIoU ↑ | AP ↑ | AP50 ↑ | AR ↑ | CLIP ↑ | Local CLIP ↑ | Intra-LPIPS ↓ | Aesthetic Score ↑ |
| w/o CPM | **0.70** | **0.24** | **0.42** | **0.48** | 31.93 | 27.51 | **0.5711** | 6.08 |
| $w = 0.1$ | 0.66 | 0.21 | 0.37 | 0.44 | **32.08** | 27.50 | 0.5783 | **6.11** |
| $w = 0.15$ | 0.68 | 0.23 | 0.40 | 0.46 | 32.02 | **27.60** | 0.5763 | **6.11** |
| $w = 0.2$ | 0.69 | **0.24** | **0.42** | 0.47 | 31.94 | 27.56 | 0.5751 | **6.11** |

### A.7    USER STUDY

We conducted a user study to evaluate the generated panoramas along four dimensions: layout fidelity, prompt consistency, style coherence, and overall visual quality.

In our user study, participants were shown several groups of images. Each group consisted of a "bounding-box reference / textual prompt" and four generated outdoor panoramic images (labeled as Image A, Image B, Image C, and Image D). Participants rated each image on four dimensions:

layout fidelity, prompt consistency, style coherence, and overall visual quality, using a scale from 1 (lowest) to 5 (highest).

Table 9 reports the average ratings for each method. As shown, our method, GAF-Pano, achieves the highest scores across all dimensions, suggesting that the panoramas it generates are more preferred by human evaluators in terms of layout, visual quality, prompt fidelity, and style consistency.

Table 9: User study results (average ratings) across the four evaluation dimensions.

| Method | Layout Fidelity | Text-Image Consistency | Stylistic Coherence | Visual Quality |
|---|---|---|---|---|
| MultiDiffusion | 3.04 | 2.49 | 3.20 | 2.75 |
| SyncDiffusion | 2.98 | 2.69 | 3.29 | 2.82 |
| MAD | 3.05 | 2.84 | 3.22 | 2.84 |
| GAF-Pano | **4.16** | **3.93** | **4.20** | **4.33** |

## A.8 The Agent for Identifying Single-Object and Multi-Object Prompts

We use a GPT-4.1 based agent to classify phrases into single-object ($S_{\text{single}}$) or multi-object ($S_{\text{multi}}$) categories, which dictates the assignment of conditional positional masks or uniform masks. To validate this agent, we evaluated it on 463 local descriptions randomly sampled from 100 panoramic layouts against manually annotated ground truth. The agent achieved 97.41% accuracy (451/463), confirming its reliability for the CPM strategy. The complete prompt used for this agent is provided below.

---

**LLM-based prompt template used by the agent for classifying phrases into $S_{\text{single}}$ or $S_{\text{multi}}$**

You are an expert in English grammar and semantic analysis. Your task is to analyze phrases and determine whether they should be treated as **SINGULAR** or **PLURAL** for image generation purposes.

**Important Rules:**
1. **PLURAL (return true):** - Multiple discrete objects: "two dogs", "three cars", "many people", "group of people" - Natural plurals: "clouds", "trees", "flowers", "birds", "buildings" - Continuous environments: "sky", "ocean", "grass", "water", "sand", "fog", "mist", "river", "lake", "forest", "mountain" - Abstract/environmental concepts: "sunlight", "atmosphere", "wind", "rain", "snow" - Landscapes/terrains: "beach", "desert", "field", "meadow"
2. **SINGULAR (return false):** - One discrete object: "a dog", "one car", "a house", "a tree" - One person/animal: "a man", "a woman", "a child", "a cat" - One item: "a chair", "a table", "a book"

**Key Point:** Environmental elements (e.g., "sky", "ocean", "mountain", "forest", "beach") are always PLURAL.

Analyze the following phrases and return ONLY a JSON array of boolean values (`true` for PLURAL, `false` for SINGULAR), nothing else.

Phrases: {phrases}

Example: [true, false, true]

---

## A.9 The Pano-Layout-Bench

As introduced in the main paper, *Pano-Layout-Bench* is established to support layout-to-image generation under panoramic settings. The benchmark was constructed through a semi-automated process: a multimodal LLM (GPT-4o Achiam et al. (2023)) generated diverse scene descriptions with bounding box layouts, which were then manually refined to ensure logical coherence and realism. We design the prompt templates in Figure 16 with instructions and in-context examples. The LLM follows the instructions to generate panoramic object layouts, which are then used as input for L2I methods to generate the final images. As shown in Table 10, the dataset includes panoramic layouts with three aspect ratios: 1:2 (412 samples), 1:3 (456 samples), and 1:4 (473 samples). On average,

each bounding box covers approximately 15.4% of the image area, with a mean width of 0.4416 and a mean height of 0.2579 (normalized to the image resolution).

Figure 11 shows that the dataset covers diverse scene types, including urban (e.g., *city*, *market*), natural (e.g., *forest*, *beach*, *mountain*), and others, providing a broad context for layout conditioning. Additionally, Figure 12 presents the top 20 most frequently occurring objects, ranging from natural elements (e.g., *trees*, *waves*, *mountains*) to man-made or animate entities (e.g., *children*, *skyscrapers*, *people*), supporting diverse object arrangement patterns for controllable image synthesis.

Table 10: Statistics of samples with different aspect ratios and overall bounding box distributions.

| Aspect Ratio | 1:2 | 1:3 | 1:4 |
|---|---|---|---|
| #Samples | 412 | 456 | 473 |

| BBox Statistic | Value |
|---|---|
| Mean Width | 0.4416 |
| Mean Height | 0.2579 |
| Mean BBox Area | 0.1540 |

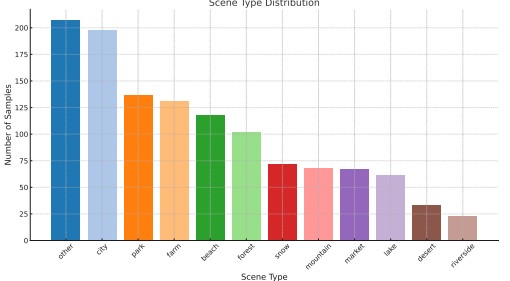 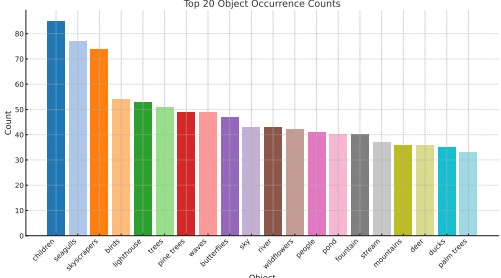

Figure 11: Scene type distribution in the Pano-Layout-Bench. The dataset covers a diverse range of scene categories such as city, park, beach, forest, and others.

Figure 12: Top 20 most frequently occurring objects in the Pano-Layout-Bench. The dataset includes a variety of natural and man-made objects, enabling diverse layout compositions.

## A.10 ADDITIONAL QUALITATIVE RESULTS

We present additional qualitative comparisons between our method and the baselines under various aspect ratios (1:2, 1:3, and 1:4) in Figure 14. All results are generated using background prompts with the short side fixed to 1024 pixels. As shown, our method produces images that better respect the specified layouts, achieving higher fidelity across diverse panoramic settings.

We also provide more results generated using our method with holistic prompts in Figure 15. All prompts are provided at the end of the appendix in the same visual order (left to right, top to bottom)

## A.11 LIMITATIONS AND FUTURE WORK

Our method has several key limitations. First, GAF-Pano is fundamentally constrained by the layout control capabilities of the underlying pre-trained layout-to-image model, meaning that any limitations in object placement or spatial reasoning from the base model will propagate to our panoramic results. Second, there exists a distributional mismatch between our evaluation setting and the training paradigm of pre-trained models. Most layout-to-image models are trained with holistic prompts containing comprehensive scene descriptions, while our fair evaluation uses background-only prompts. This mismatch can lead to incomplete object generation or missing elements, as evidenced by the performance gap between our background-only and holistic prompt results (GAF-Pano vs GAF-Pano*). CPM may also result in missing content generation if the bounding boxes are small. Additionally, the global attention fusion mechanism introduces computational overhead during inference, and may not capture the most nuanced cross-view dependencies for complex multi-view objects. Moreover, the plausibility of the provided layouts also affects the final generation quality. Figure 13 shows some failure cases.

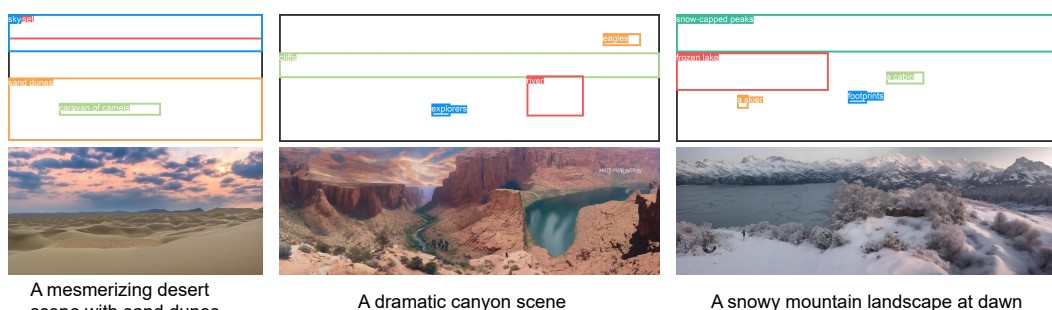

A mesmerizing desert scene with sand dunes

A dramatic canyon scene

A snowy mountain landscape at dawn

Figure 13: Some failure cases of our methods.

As future work, we plan to investigate more efficient attention mechanisms, such as the linear attention proposed in Sana Xie et al. (2024), to reduce the computational overhead introduced by global attention fusion. In addition, extending layout-to-image models with explicit background-focused training may help address the distributional mismatch observed in our evaluation and further validate the effectiveness of our framework.

### A.12 THE USE OF LARGE LANGUAGE MODELS (LLMS)

We employed large language models (Gemini and ChatGPT) in limited ways to support our research and writing. Specifically:

- Writing polish: For example, we provided experimental tables and our own manual analysis, and asked the model to help rephrase the text while respecting the actual results. The authors then carefully reviewed and revised the suggestions to ensure accuracy and appropriateness, resulting in the final version presented in the paper.
- Experimental implementation assistance: We used LLMs to assist in implementing parts of the experimental code. All generated code was verified and, where necessary, modified by the authors to ensure correctness.
- Technical formatting: We used LLMs for routine tasks such as generating LaTeX table code from our manually prepared experimental data. Again, the authors verified all generated content.

The authors remain fully responsible for the correctness and originality of all content.

### HOLISTIC PROMPTS USED IN FIGURE 15

Below we list all the holistic prompts used to generate the results shown in Figure 15. The prompts correspond to the images in the figure in row-wise reading order (left to right, top to bottom).

- Prompt 1: A peaceful ocean view from a cliff with waves crashing against the rocks, a lighthouse in the distance, seagulls flying around, and the sun setting on the horizon.
- Prompt 2: A dramatic canyon scene with red sandstone cliffs, a river snaking through, and hikers exploring the rocky terrain.
- Prompt 3: A vibrant coral reef under the sea with colorful fish, a sea turtle swimming, and sun rays filtering through the water.
- Prompt 4: Cozy snowy holiday village at dusk, gentle snowfall, warm window lights, houses with snowy roofs, holiday market stalls with lights, people playing, cinematic warm glow, high detail.
- Prompt 5: A serene mountain landscape with high cliffs, pine forests covering the slopes, hikers reaching an overlook, a river winding through the valley, and a clear azure sky.
- Prompt 6: A cozy wooden cabin with stone chimney in snowy mountains at winter twilight with warm glowing windows, pine trees, snow-covered peaks.
- Prompt 7: A futuristic cityscape with a skyline filled with sleek skyscrapers, flying cars zooming between buildings, neon lights illuminating the scene, people in futuristic attire walking along elevated walkways, and digital billboards flashing advertisements.
- Prompt 8: A magical floating island in the sky with a castle on top, waterfalls cascading from its edges down into the clouds below, during a soft sunrise.

- Prompt 9: Majestic waterfall cascading down rocky cliffs, lush vegetation on the sides, and people standing on an observation deck admiring the view.

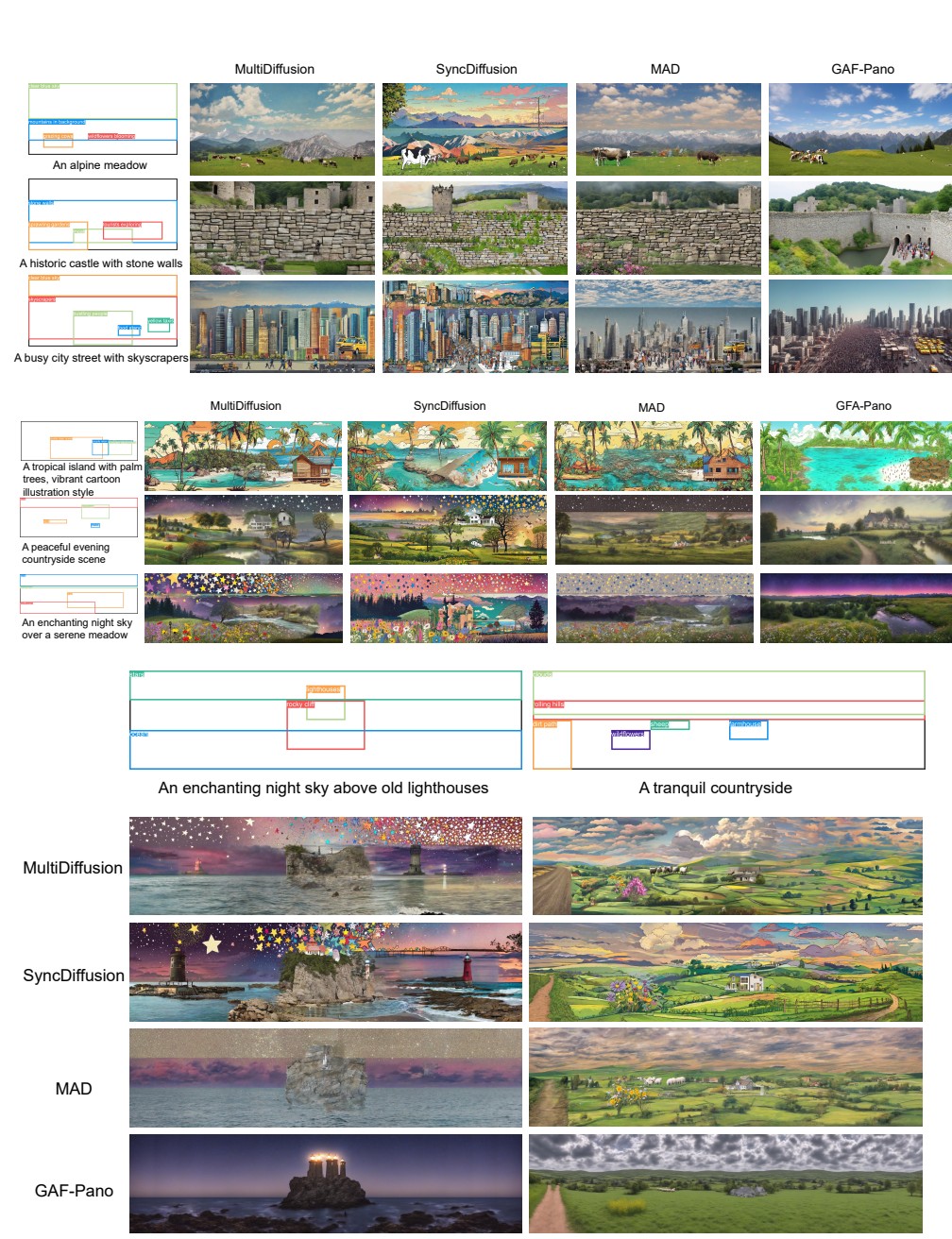

Figure 14: Additional qualitative results comparing our method with the baselines using background prompts on panoramic images with aspect ratios of 1:2, 1:3, and 1:4 (from top to bottom). All examples are generated with a fixed short side of 1024 pixels and are zoomed-in for better viewing.

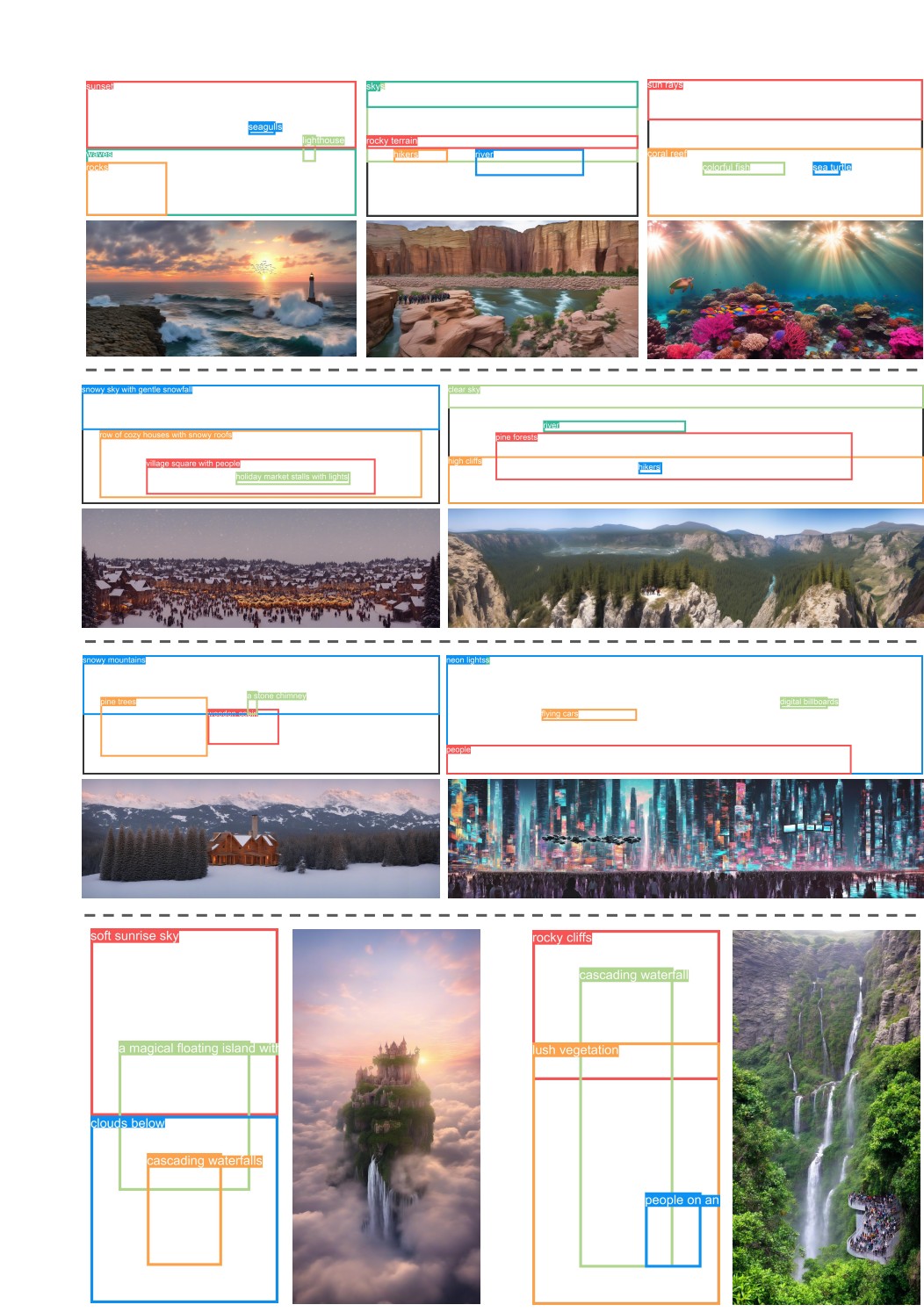

Figure 15: More results generated using our method with holistic prompts.

**Instruction**

You are a creative AI tasked with generating image descriptions for a dataset.
For each image, you will provide a detailed scene description, brief description, a list of objects in the scene,
their corresponding bounding boxes, and the aspect ratio of the image. The image should follow the "1:2" aspect ratio.

For each scene, follow these guidelines:
1. **Scene Description**: Provide a detailed description of the scene, including the objects, background, and other visual elements.
2. **Background Prompt**: Provide a concise description of the scene, focusing primarily on the background with fewer objects.
3. **Objects (Phrases)**: List at least 3 and at most 10 objects or elements in the scene. These can include people, animals, landscapes, buildings, etc.
4. **Bounding Boxes**: For each object, generate a bounding box in the format [xmin, ymin, xmax, ymax], where each value is between 0 and 1, indicating the relative position of the object in the image.
5. **Aspect Ratio**: The aspect ratio should be "1:2" (in pixels: 1024 x 2048). The width of the image is twice its height. Ensure the objects and their positions are appropriately scaled and placed within this aspect ratio.

Please ensure:
- The bounding boxes are proportional to the sizes of the objects.
- Objects should be logically placed within the scene (e.g., trees should be at the bottom, a person should be positioned naturally).
- The bounding boxes for smaller objects should not be excessively large.
- The aspect ratio must be maintained.

**Examples**

Here are some examples to guide your generation:
```
[
  {
    "id": 1,
    "prompt": "A peaceful seaside scene with a beach, ocean waves, palm trees, sunset sky, and a person with a corgi walking by the shore",
    "background_prompt": "A seaside scene",
    "phrases": ["beach", "ocean", "palm trees", "sunset sky", "a person walking", "a corgi"],
    "boxes": [[0.0, 0.6, 1.0, 1.0], [0.0, 0.3, 1.0, 0.7], [0.1, 0.1, 0.3, 0.6], [0.0, 0.0, 1.0, 0.3], [0.7, 0.4, 0.85, 0.95], [0.5, 0.6, 0.6, 0.8]],
    "aspect_ratio": "1:2"
  },
  {
    "id": 2,
    "prompt": "A beautiful landscape with mountains, trees, and a cat",
    "prompt": "A natural landscape",
    "phrases": ["mountains", "a cat", "trees", "sky with sunset"],
    "boxes": [[0.0, 0.2, 1.0, 1.0], [0.1, 0.6, 0.4, 0.8], [0.6, 0.2, 0.9, 0.8], [0.0, 0.0, 1.0, 0.3]],
    "aspect_ratio": "1:2"
  }
]
```

Now, please generate a batch of 25 responses with the same JSON format.
I will parse your response using regex, like json_pattern = r'```json\n(.*?)\n```', so please ensure your response is in valid JSON format and enclosed in a ```json code block```.

**Output**
**prompt**:"An ethereal panoramic view of an enchanted autumn valley at twilight. A meandering river of liquid starlight flows through a forest of colossal, ancient trees with bioluminescent patterns on their bark. Luminous mist swirls around their bases. In the far distance, majestic mountains with crystal-tipped peaks pierce a celestial nebula sky filled with shimmering constellations."
**phrases**:[ "celestial nebula sky", "crystal-tipped mountains", "colossal enchanted trees", "glowing river of starlight", "luminous swirling mist" ]
**boxes**:[[0.0, 0.0, 1.0, 0.4], [0.0, 0.25, 1.0, 0.5], [0.0, 0.3, 1.0, 0.8], [0.1, 0.6, 0.9, 1.0], [0.0, 0.5, 1.0, 0.9]]

**ChatGPT**

Figure 16: The instructions for ChatGPT to generate the layout for the panoramic images.

