# OpenReview forum: "GAF-Pano: Zero-Shot Layout-Controlled Panorama Generation via Global Attention Fusion"
_ICLR.cc/2026/Conference — ICLR 2026 Conference Withdrawn Submission_

### Official Review · Reviewer_ogCv · 2025-10-29

**Soundness:** 2
**Presentation:** 3
**Contribution:** 2
**Rating:** 4
**Confidence:** 4

**Summary:**

The paper proposes a method for layout-conditioned panorama (wide aspect ratio images). It presents GAF-Pano, a training-free framework, that augments layout-to-image (L2I) diffusion models using a "Global Attention Fusion (GAF)" mechanism then enables semantic coherence of disjoint views during inference as well as improved spatial control. The methods builds on previous work that introduced similar fusion mechanisms for this task of layout-controlled generation, and shows improved quantitative and qualitative results.

**Strengths:**

Strengths
- Well designed pipeline and training-free framework. The qualitative results are interesting and show improvement over baselines.
- Qualitative results show good coherence
- Benchmark (Pano Layout Bench) has value to the community

**Weaknesses:**

- The initial background/introduction of attention fusion is limited, and I found myself forced to re-read numerous times before properly grasping the concept. This is not a standard algorithm that readers should be expected to know, and greater care to improving the writing here should be made.
- The “theoretical foundation” claimed in Section 3 is entirely qualitative. No mathematical analysis or justification is given, is it missing?
- The contribution is somewhat limited, and primarily hinges on applying the fusion mechanism from [1]. The CPM is a key practical contribution, and therefore should be presented with importance in the main paper. I see the ablation in the supp, however I do not agree with the analysis: "As illustrated in Figure 5 and confirmed by the quantitative results in Table 6, introducing CPM reduces object duplication and visual artifacts compared to the baseline (w/o CPM), while maintaining competitive layout fidelity and text-image consistency." While Figure 5 does show less artifacts/duplication, it is unclear how cherry picked these two results are in the larger benchmark, and Table 6 shows w/o CPM has better performance across the board, bringing into question the value of this contribution.
- The description of baseline implementation is lacking. Authors reference Appendix A.1 for more details, however it is unclear how MAD is implemented, and how layout control is implemented for it.
- The contribution of the benchmark is downplayed, moving details to the appendix, when it remains a main contribution of the work overall

[1] Quattrini, Fabio, et al. "Merging and splitting diffusion paths for semantically coherent panoramas." European Conference on Computer Vision. Cham: Springer Nature Switzerland, 2024.

**Questions:**

- My main question is to the value of CPM and the differentiation of this work to MAD. The contribution does not seem sufficiently novel and the value of CPM seems weak.

---

> ### Author Response · Authors · 2025-11-21
>
> We thank the reviewer for recognizing the value of our pipeline, the quality of our results, and the contribution of Pano-Layout-Bench. We appreciate the constructive feedback regarding clarity. We address your concerns below, particularly clarifying the differentiation from MAD and analyzing the trade-offs involved in CPM.
>
> **1. Differentiation from MAD and Novelty (Response to Weakness 3, 4 & Question 1)**
>
> The distinction between GAF-Pano and MAD (Quattrini et al., 2024) lies in the **Integration Target** and the **Mechanism of Layout Control**, which leads to superior "Layout Scalability."
>
> - **Targeting L2I vs. Standard T2I:**
>
> Existing methods like MAD primarily operate as "Plug-and-Play" modules to MultiDiffusion. When applied to region-based control (MultiDiffusion Application 2), they generate regions independently and fuse them via latent averaging using defined regional masks.
>
> In contrast, GAF-Pano is the first to integrate Global Attention Fusion specifically into **Layout-to-Image (L2I)** models. We do not just blend overlapping views; we fuse the specific attention layers responsible for layout guidance.
>
> - **Improved Layout Scalability:**
>
> **Improved Layout Scalability:** While MAD utilizes attention fusion for *semantic scalability* (e.g., semantic coherence), GAF-Pano demonstrates significantly improved **"Layout Scalability."** With the defined region masks, MAD first generates object in local region separately. Although the generation of objects **within each region** acts coherently, the lack of unified global layout guidance during the averaging stage often leads to misalignment and fragmentation artifacts at region boundaries. **GAF-Pano overcomes this via "Holistic Layout Planning."** By synchronizing the Layout-Guided Cross-Attention layers globally, our model "perceives" a large, cross-view bounding box as a single continuous entity, ensuring consistent object generation across views.
>
> **2. The Value of CPM and Table 6 (Response to Weakness 3 & Question 1)**
>
> You are correct that `w/o CPM` achieves higher Layout Fidelity scores (mIoU). However, we observe that this is a case where **standard metrics diverge from perceptual semantic correctness**.
>
> - **The "Fidelity vs. Repetition" Trade-off:**
>
> Without CPM, the model tends to maximize attention scores broadly to satisfy the prompt. In large boxes, this often results in **object repetition** (e.g., generating two cats side-by-side instead of one large cat).
>
> **Why metrics favor `w/o CPM`:** Metrics like mIoU reward filling the box with "object pixels." Generating two cats fills the box more effectively than one cat with negative space, yielding higher mIoU despite the semantic error.
>
> **Why `w/ CPM` is better:** CPM enforces a single focal point. This inevitably reduces the total "fill rate" (lowering mIoU) but corrects the semantic error of duplication.
>
> - **New Manual Evaluation:**
>
> To validate this beyond automated metrics, we manually evaluated **362 images** comparing single-object generation.
>
> **Success (CPM reduced artifacts):** 41.7% (75 cases)
>
> **both success:** 24.4% (44 cases)
>
> **Failure:** 33.9% (61 cases)
>
> The fact that CPM successfully reduced artifacts in nearly **42%** of cases (zero-shot) demonstrates its practical value. We believe generating the correct quantity of objects is more important for users than maximizing pixel coverage, even if metrics penalize it.
>
> **3. Theoretical Foundation vs. Mechanism Analysis (Response to Weakness 2)**
>
> Thank you for the correction.  The use of "theoretical foundation" was a typo in our previous draft. We have revised it to **"empirical motivation"** in the updated paper to properly describe our qualitative analysis.
>
> **4. Clarity of Attention Fusion (Response to Weakness 1)**
>
> We have rewritten the introduction to explicitly define Attention Fusion early on as a mechanism aggregating Query/Key/Value matrices into a shared global context.
>
> **5. Baseline Implementation Details (Response to Weakness 4)**
>
> We have updated the paper to explicitly state that MAD was evaluated using its official codebase in "Region-Controlled" mode. This clarifies that its layout control relies on MultiDiffusion region based generation, distinguishing it from our L2I-integrated approach.
>
> **6. Benchmark Contribution (Response to Weakness 5)**
>
> We agree with the reviewer that the benchmark is a main contribution and should be highlighted in the main text.
>
> In the revised Section 6.1, we have integrated key construction details and comprehensive statistics that were previously in the appendix.
>
> We believe these additions provide a clear and immediate view of the benchmark's quality and scope within the main paper.

---

### Official Review · Reviewer_aV6y · 2025-10-29

**Soundness:** 3
**Presentation:** 3
**Contribution:** 2
**Rating:** 4
**Confidence:** 4

**Summary:**

This paper proposed GAF-Pano, a training-free framework for zero-shot layout-controlled panorama generation. The framework integrates a Global Attention Fusion mechanism into a pre-trained layout-to-image model, using a Global Context Synchronization, Fusion, and Dispatch workflow (SFD) operating within the attention layers of the diffusion model. GAF-Pano aims to overcome the limitations of prior panorama generation methods in terms of global semantic coherence and fine-grained layout control. Ablation experiments and performance comparisons verify its effectiveness.

**Strengths:**

1. This paper proposes a novel method that integrates the global attention fusion mechanism into a pre-trained layout-to-image model in a training-free manner, achieving high-precision control of panorama generation and further completing a wider range of tasks.

2. Figures like Figures 2 and 3 are very cleverly chosen to illustrate the benefits of attention fusion and the motivation for this paper.

3. Detailed ablation results are provided in the main text and supplementary materials, deeply analyzing the contributions of self-attention fusion, cross-attention and conditional position mask, and confirming the rationality of the method.

**Weaknesses:**

1. Lack of recent key papers on panoramic image generation, such as PanFusion (Zhang et al., 2024) and DiT360 (Feng et al., 2025) — they are not cited or explicitly discussed. These works pursue very similar goals (maintaining semantic coherence and layout control in panoramic synthesis)

2. The paper shows failure cases (Figure 11), but lacks quantitative error analysis, e.g., how often certain types of failures occur, or how they correlate with prompt/box complexity. This would put the robustness of the approach into context.

3. The method is tailored and benchmarked for wide-aspect-ratio panorama generation. It is uncertain whether the core global fusion principle can be generalized to more arbitrary spatial arrangements, such as 360-degree panoramas.

**Questions:**

1. Can the authors provide experimental comparisons with PanFusion or DiT360?

2. For scenes with many objects (dense layouts), does this method scale well in terms of computational cost and consistency, or does performance degrade dramatically?

---

> ### Author Response · Authors · 2025-11-23
>
> **1. Efficiency and Scalability Analysis (Response to Weakness 2 & Question 2)**
>
> We appreciate the reviewer’s question regarding the scalability and consistency of our method in dense layout scenarios. In response, we have conducted a comprehensive evaluation of computational cost and generation stability across varying layout complexities. These new experiments and a detailed analysis have been updated in Appendix A.3 of the revised paper.
>
> **Computational Scalability ($O(1)$ vs. $O(N)$)**
>
> As detailed in **Table 1**, our analysis reveals a critical advantage of GAF-Pano over existing region-based methods. Baselines like MultiDiffusion, MAD, and SyncDiffusion exhibit a linear increase in inference time as the number of bounding boxes ($N$) grows. For instance, MultiDiffusion suffers a 281% slowdown when scaling from 2 to 10 boxes.
>
> In contrast, GAF-Pano maintains an effectively constant inference speed ($O(1)$). Our inference time increases negligibly by only 1.9% (from 63.6s to 64.8s) as $N$ increases to 10. This efficiency is achieved because our global context fusion processes all layout conditions in parallel, avoiding the bottleneck of sequential or separate region generation.
>
> Table 1. Comparison of Inference Time (seconds) with Varying Layout Complexity.
>
> | **Method**          | **N=2**  | **N=6**  | **N=10** | **Relative Increase (%, N=2→10)** |
> | ------------------- | -------- | -------- | -------- | --------------------------------- |
> | MultiDiffusion      | **59.2** | 142.5    | 225.5    | +281.0%                           |
> | MAD                 | 70.9     | 160.4    | 249.7    | +252.1%                           |
> | SyncDiffusion       | 111.6    | 193.8    | 277.6    | +148.7%                           |
> | **GAF-Pano (Ours)** | 63.6     | **63.8** | **64.8** | **+1.9%**                         |
>
> **Performance Stability (Success Rate)**
>
> To evaluate robustness, we analyzed the **Performance Degradation** trend as scene complexity increases. We measured the Success Rate @ 50 (SR@50) and Average IoU.
>
> As shown in **Table 2**, GAF-Pano demonstrates remarkable stability. While there is a slight natural variance as complexity grows, the model maintains a high precision (SR@50 > 0.83) and steady IoU (~0.71) even when handling up to 8 concurrent phrases. This confirms that our method scales effectively without the dramatic performance degradation often seen in dense layouts.
>
> Table 2. Performance Stability Analysis.
>
> Success Rate (SR@50) and Average IoU remain stable as the number of layout phrases increases.
>
> | **Num Phrases** | **Match Rate (SR@50)** | **Average IoU** | **Sample Count** |
> | --------------- | ---------------------- | --------------- | ---------------- |
> | 2               | 0.9333                 | 0.7856          | 15               |
> | 3               | 0.8608                 | 0.7056          | 79               |
> | 4               | 0.8607                 | 0.7082          | 359              |
> | 5               | 0.8570                 | 0.7117          | 544              |
> | 6               | 0.8419                 | 0.7082          | 311              |
> | 7               | 0.8341                 | 0.6976          | 31               |
> | 8               | 0.8750                 | 0.6804          | 2                |

---

> ### Author Response · Authors · 2025-11-27
>
> 1. **Comparison with PanFusion and DiT360 (Response to Weakness 1 & Question 1)**
>
> We thank the reviewer for pointing out PanFusion[1] and DiT360[2]. We acknowledge their contributions to 360° panorama generation and have discussed them in our revised "Related Work."
>
> However, we respectfully clarify that these works are not suitable baselines for our method due to fundamental differences in **geometric definition** and **control objectives**:
>
> - **Spherical vs. Planar Wide-Aspect-Ratio Formulation:** PanFusion and DiT360 focus on 360° spherical panoramas (typically via equirectangular projection), addressing specific geometric challenges like pole distortion and loop closure. In contrast, our work strictly scopes "panorama" as **wide-aspect-ratio images generated through horizontal extension and view stitching**(Following MultiDiffusion, SyncDiffusion and MAD). Our method operates on an extended 2D planar canvas without spherical constraints. Our task operates in a planar coordinate system without spherical constraints, making direct quantitative comparison geometrically invalid.
>
> - **Structural vs. Instance-Level Control:** The layout control in PanFusion relies on global distance/depth maps via ControlNet to define 3D room structures (e.g., walls, ceilings), as evidenced by the room layout renderings in the supplementary material of [1] (see Figure B.1). We acknowledge that existing joint diffusion frameworks, such as MultiDiffusion equipped with ControlNet (Depth/Canny) or SyncTweedies[3] (which utilizes depth-driven synchronized mechanisms), can already achieve similar global geometric consistency. Unlike global structural conditioning, our framework targets **fine-grained, bounding-box-level Layout-to-Image (L2I) generation**. Given a tuple $T_{LCPG}=(P,\{B,D\})$ consisting of bounding boxes and local prompts, our goal is to ensure precise semantic object grounding (e.g., generating "a cat" in box A and "a vase" in box B with distinct attributes). This **instance-level semantic control** differs fundamentally from the global structural layout control found in PanFusion or depth-driven methods like SyncTweedies.
>
> In summary, while PanFusion and DiT360 excel at spherical structure, they do not address the explicit, bounding-box-based instance control for planar wide images that is central to our contribution.
>
> 2. **Generalizability to 360-Degree Panoramas (Response to Weakness 3)**
>
> We appreciate the reviewer regarding the generalization of our core fusion principle to 360-degree spatial arrangements. We would like to clarify the positioning of our method relative to these global architectures and discuss the potential applicability of our proposed mechanism to other MultiDiffusion-based frameworks.
>
> **Distinct from Global Architectures**: Methods like DiT360 and PanFusion inherently achieve global coherence through the global receptive field of Transformers or specialized dual-branch cross-attention mechanism (e.g., EPPA modules). Thus, they **do not require the view-based fusion strategy** used in our method.
>
> However, it is worth noting that UNet-based MultiDiffusion frameworks remain relevant for 360° panorama generation, exemplified by **StitchDiffusion**[4]. Since StitchDiffusion essentially incorporates a "Stitch Block" strategy into the MultiDiffusion denoising path to address seam continuity , it appears architecturally compatible with the Attention Fusion mechanism and the Sync-Fuse-Dispatch (SFD) workflow. This can serve as a valuable direction for further exploration.
>
> [1] Zhang et al. Taming Stable Diffusion for Text to 360 Panorama Image Generation, CVPR 2024.
>
> [2] Feng et al. DiT360: High-Fidelity Panoramic Image Generation via Hybrid Training, arXiv 2025.
>
> [3] Kim et al. SyncTweedies: A General Generative Framework Based on Synchronized Diffusions, NeurIPS 2024.
>
> [4] Wang et al. Customizing 360-Degree Panoramas through Text-to-Image Diffusion Models, WACV 2024.

---

### Official Review · Reviewer_4Stg · 2025-10-31

**Soundness:** 3
**Presentation:** 3
**Contribution:** 3
**Rating:** 4
**Confidence:** 5

**Summary:**

This paper addresses the dual challenges of achieving global semantic coherence and precise local layout control in the generation of wide-aspect-ratio panoramic images. The authors identify that existing methods, which typically synchronize independent views, often suffer from object fragmentation and contextual artifacts, failing to provide fine-grained control over object placement.
To solve this, the authors propose **GAF-Pano, a training-free, zero-shot framework**. The core innovation is the integration of a Global Attention Fusion mechanism into a pre-trained Layout-to-Image (L2I) model. The authors claim this mechanism activates a latent "Global Semantic Modeling Capability" within the model, enabling holistic planning across the entire panoramic canvas.
This is implemented through a systematic **Sync-Fuse-Dispatch (SFD) workflow** operating within the attention layers at each diffusion sampling step. The workflow consists of three stages: (1) Sync: Aggregates latent features from all local views into a unified global context. (2) Fuse: Performs multi-level attention (self-attention for structure, cross-attention for semantics and layout) on this global context to merge information across views. (3) Dispatch: Propagates the globally-enriched features back to the individual local view pipelines.
Additionally, the paper introduces a **Conditional Position Mask (CPM)** to mitigate object repetition within large bounding boxes. To rigorously evaluate their method, the authors constructed a new benchmark, **Pano-Layout-Bench**. Experimental results show that GAF-Pano significantly outperforms strong baselines like MultiDiffusion, SyncDiffusion, and MAD across various metrics, including layout fidelity (mIoU, AP), text-image consistency (CLIP Score), and style coherence (Intra-LPIPS).

**Strengths:**

- **Fundamental breakthrough on the scaling problem**: The paper tackles a core bottleneck in panorama generation: the memory limitation that arises from scaling. As noted in the pre-experiment in Section 3, directly applying modern L2I models to wide-aspect-ratio canvases is practically infeasible due to prohibitive memory requirements. Existing joint diffusion methods like MultiDiffusion adopted a 'divide-and-conquer' approach by splitting the panorama into overlapping views. However, this workaround introduced a critical side effect of "information fragmentation," leading to object segmentation and semantic inconsistencies.
The primary strength of this paper is its precise diagnosis of this problem and its proposal of a fundamental solution that 'repairs' the existing paradigm. Instead of merely stitching views, GAF-Pano introduces **Global Attention Fusion**, which aggregates latent features from all views into a single global context before the attention operation. This overcomes the limitations of previous methods that reasoned independently on fragmented information, enabling the model to perform true "holistic planning" within a memory-efficient framework. This paradigm shift is the key contribution that unlocks both global coherence and precise local control at scale. From this perspective, the problem addressed is not just about layout control but the more fundamental challenge of scalability in generative models. The abstract would be more impactful if it mentioned this memory constraint before discussing global coherence.

- **Step-by-step method**: The Sync-Fuse-Dispatch (SFD) workflow that implements this idea is logically structured into information aggregation (Sync), global reasoning (Fuse), and information propagation (Dispatch). Each component has a clear role and is explained in a step-by-step manner, making the methodology easy for readers to understand.

- **Experimental Validation**: Authors have conducted an impressive suite of experiments to validate their method's performance. Recognizing the lack of a standardized evaluation protocol, they constructed a new benchmark, Pano-Layout-Bench. According to Appendix A.7, this benchmark consists of 1,341 unique prompts across three aspect ratios (1:2, ... 1:4) and includes a diverse range of scene types and object distributions. The evaluation is comprehensive, using metrics across four distinct categories: Layout Fidelity (mIoU, AP), Text-Image Consistency (CLIP), Stylistic Coherence (Intra-LPIPS), and Visual Quality (Aesthetic Score). The User Study detailed in Appendix A.6 is particularly persuasive. In human evaluations, GAF-Pano overwhelmingly outperformed competing models across all dimensions, including layout fidelity, prompt consistency, style coherence, and overall visual quality.

**Weaknesses:**

- The 'Related Work' is deficient in both structure and content. First, its placement in Section 5, rather than immediately after the introduction, disrupts the logical flow. Second, its content is misaligned with the paper's core topic. While the introduction correctly identifies MultiDiffusion, SyncDiffusion, GVCFDiffusion, PanoFree, and MAD as key prior works for comparison, the 'Related Work' section fails to provide any in-depth discussion of SyncDiffusion, PanoFree, or MAD. Instead, it offers a superficial list of general L2I models like ControlNet and GLIGEN. This omission makes it impossible for the reader to understand the novelty of GAF-Pano in the context of existing panorama generation research.
- It's hard to grasp the framework flow when looking at core mechanism pipeline, in Fig. 2 and Fig. 4.
- There are concerns about the implementation of the SyncDiffusion baseline. The main text refers to Appendix A.1 for details on how baselines were adapted for layout control, but the explanation in Appendix A.1.2 is superficial. The authors state in the introduction that SyncDiffusion uses a perceptual loss for smooth transitions, yet there is no mention of how this core mechanism was applied for layout control in the appendix. Furthermore, since SyncDiffusion is not inherently an L2I method, the lack of a specific explanation for how bounding boxes and object descriptions were integrated severely harms reproducibility.
- Table 1 shows that GAF-Pano's performance improves substantially when using holistic prompts compared to background-only prompts. While Appendix A.2 and Fig.8 demonstrate that MultiDiffusion suffers from object duplication and fragmentation issues due to prompt leakage when using holistic prompts, there is insufficient explanation of why GAF-Pano is more robust to these problems.
- The authors present GAF-Pano as a general framework applicable to pre-trained L2I models, but experiments are largely limited to a single model—SDXL-based IFAdapter. Even the limitations section in Appendix A.9 acknowledges that "GAF-Pano is fundamentally constrained by the layout control capabilities of the underlying pre-trained layout-to-image model." While Figure 7 in Appendix A.1.1 qualitatively shows that GAF-Pano resolves direct generation failures across three different L2I models (InstanceDiffusion, IFAdapter, CreatiLayout), without quantitative comparison.
- Appendix A.3 provides overall inference times and lacks analysis of memory overhead or layer-wise computational costs. In the 'Sync' step, which aggregates a global context, could potentially reintroduce a memory bottleneck.

**Questions:**

- The SyncDiffusion baseline in Table 1 shows particularly low performance. Could you please provide the specific implementation details missing from Appendix A.1.2, particularly how the core 'perceptual loss' mechanism of SyncDiffusion was adapted for the layout control task?
- Why does 'Related Work', not to discuss the key prior works (e.g., SyncDiffusion, MAD) that were mentioned in the introduction?
- Is it possible to analyze the computational cost of each step in the S-F-D workflow?
- The Conditional Position Mask (CPM) proposed in Section 4.3.2 depends on an **agent** for prompt classification. Could you please specify the mechanism of this agent (LLM-based?) and its classification accuracy on dataset?
- Would you provide quantitative comparisons for GAF-Pano when integrated with different backbone models?
- In Eqn 13, $SPM_i$ is a typo for $CPM_i$ ?

---

> ### Author Response · Authors · 2025-11-21
>
> We thank the reviewer for their thoughtful and constructive feedback. We are encouraged by your recognition of our work as a "fundamental breakthrough on the scaling problem" and your appreciation of our "precise diagnosis" of the fragmentation issues in existing methods. Below, we address your questions and concerns point-by-point.
>
> **1. Implementation of SyncDiffusion Baseline (Response to Weakness 3 & Question 1)**
>
> We clarify that SyncDiffusion was originally proposed as a "plug-and-play module" to enhance the global and stylistic coherence of panoramas via a perceptual similarity loss, rather than being a method designed specifically for layout control.
>
> **Implementation Details:**
> Given the "plug-and-play" nature of the original SyncDiffusion, it remains directly applicable to the **MultiDiffusion framework's region-based generation** setting. We strictly followed the methodology described in that application:
>
> - **Layout Mechanism:** We adopted the **region-based generation pipeline from the MultiDiffusion framework**. This follows its standard protocol of defining local masks for regions (bounding boxes) and fusing latent updates to handle spatial arrangement.
> - **SyncDiffusion Module:**  On top of this, we apply the SyncDiffusion perceptual loss. At each denoising step, we select a reference view's latent representation. We then compute the gradient of the perceptual loss between other views and this anchor. This gradient serves as guidance to enforce stronger style consistency across different views.
>
> The quantitative results in Table 1 suggest a potential trade-off in design objectives. SyncDiffusion prioritizes minimizing perceptual distance between views to ensure seamless blending. While the results confirm its effectiveness for stylistic and semantic coherence, we observe that this mechanism may unintentionally compromise the strict boundary adherence required for high-precision layout control. We have added this analysis to **Appendix A.1.2**.
>
> **2. Related Work Structure and Content (Response to Weakness 1 & Question 2)**
>
> We agree that the original Related Work section was misplaced and lacked sufficient discussion on panorama-specific baselines.
>
> - we have moved the **Related Work** section to **Section 2**, immediately following the Introduction, to ensure better logical flow.
> - Regarding the omission of works like SyncDiffusion and MAD: We initially focused on explicit layout control mechanisms. We have now removed the generic "Controllable Text-to-Image Generation" subsection with a specific **"Panorama Generation"** section. This explicitly differentiates GAF-Pano from baselines like SyncDiffusion and MAD, highlighting that while they focus primarily on semantic coherence, our method uniquely integrates global spatial planning for layout generation, which these prior works do not address.
>
> **3. Computational Cost and Memory Analysis (Response to Weakness 5 & Question 3)**
>
> We have conducted a detailed memory profiling of the **Sync-Fuse-Dispatch (S-F-D)** workflow. The quantitative results are presented below, with a corresponding visualization added to the Appendix.
>
> **Table: Peak Memory Overhead (MB) across different resolution layers**
>
> | Layer (Resolution)          | Sync   | Fuse  | Dispatch |
> | --------------------------- | ------ | ----- | -------- |
> | Down Block ($64\times256$)  | 660 MB | 80 MB | 260 MB   |
> | Down Block ($32\times128$)) | 330 MB | 40 MB | 130 MB   |
> | Mid Block ($32\times128$)   | 330 MB | 40 MB | 130 MB   |
> | Up Block ($32\times128$)    | 330 MB | 40 MB | 130 MB   |
> | Up Block ($64\times256$)    | 660 MB | 80 MB | 260 MB   |
>
> **Analysis of the Workflow:**
>
> We define Peak Memory Overhead as the maximum transient memory increase observed during a specific stage relative to the pre-stage memory state.
> - **Sync as the Bottleneck:** The **Sync** stage is the most memory-intensive due to the double buffering required to retain multi-view tensors while allocating the global aggregation canvas.
> - **Fuse (Attention) Efficiency:**  The Fuse stage incurs 40MB–80MB of overhead. Although attention retains its $O(N^2))$ computational complexity, Flash Attention avoids materializing the full $N^2$ score matrix and thus reduces the **memory footprint** to near-linear $O(N)$. This confirms that the main overhead arises from the Sync stage rather than from attention itself.
>
> **4. Conditional Position Mask (CPM) Agent Details (Response to Question 4)**
>
> - Implemented with **GPT-4.1**, the agent classifies bounding box prompts into single-object ($S_{single}$) or multi-object ($S_{multi}$) categories to assign focused or uniform masks, respectively. The detailed prompt is added to **Appendix A.8**.
> - we randomly sampled **100 panoramic layouts** (463 local descriptions) from our benchmark and manually annotated the ground truth. The agent achieved **97.41% accuracy** (451/463 correct predictions), confirming its reliability for the CPM strategy.

---

> ### Author Response · Authors · 2025-11-27
>
> **1. Robustness to Holistic Prompts (Response to Weakness 4)**
>
> The explanation lies in the fundamental difference in attention scope:
>
> - **MultiDiffusion's Limitation:** MultiDiffusion's generation is confined locally within independent views. When a holistic prompt (containing specific object descriptions) is applied, the model lacks global context and blindly attempts to generate the described objects within every local view. This leads to the "prompt leakage" and object duplication observed in **Appendix A.2**.
> - **GAF-Pano's Solution:** In contrast, GAF-Pano performs attention fusion on the **entire global context**. Crucially, we utilize **Layout-Guided Cross-Attention (LCA)** with spatial masks. This mechanism explicitly restricts the influence of object descriptions to their designated bounding boxes, effectively "shielding" the background from object tokens in the holistic prompt. This ensures that even with rich global text conditions, objects are only generated within their specified layouts.
>
> **2. Generalization to Other Models using Masked Cross-Attention (Response to Weakness 5 Question 5)**
>
> To demonstrate the versatility of our framework, we integrated GAF-Pano with MIGC [1]. We selected this method specifically because, like the IFAdapter used in our main text, **MIGC relies on layout-guided masked cross-attention for spatial control.**
>
> This experiment confirms that GAF-Pano is not limited to a specific model instance but is broadly compatible with the **attention-masking paradigm**. By applying our "Sync-Fuse-Dispatch" workflow, we show that local attention constraints can be effectively extended to a global panoramic context.
>
> The quantitative results are presented below:
>
> | **Method**      | **mIoU ↑** | **AP ↑** | **AP50 ↑** | **AR ↑** | **CLIP ↑** | **Intra-LPIPS ↓** | **Aesthetic ↑** |
> | --------------- | ---------- | -------- | ---------- | -------- | ---------- | ----------------- | --------------- |
> | GAF-Pano (MIGC) | 0.63       | 0.31     | 0.49       | 0.49     | 29.55      | 0.5133            | 5.45            |
>
> - The robust Layout Fidelity scores (mIoU 0.63, AP50 0.49) indicate that the layout masked cross-attention mechanism functions correctly within our global latent context. This proves that GAF-Pano successfully synchronizes attention masks across views, regardless of the underlying backbone (validating on SD1.5 architecture).
> - The low Intra-LPIPS score (0.5133) confirms that our global attention fusion effectively maintains stylistic consistency, further validating the robustness of the framework.
>
> We have added these results and the corresponding analysis to **Appendix A.4** to demonstrate the framework's generalization to methods sharing this control mechanism.
>
> **3. quantitative comparison with direct generation (Response to Weakness 5)**
>
> We conducted a comprehensive evaluation comparing **GAF-Pano** against the direct inference of three Layout-to-Image (L2I) models: **InstanceDiffusion**, **IFAdapter**, and **CreatiLayout**.
>
> The quantitative results are presented below:
>
> | **Method**          | **mIoU ↑** | **AP ↑** | **AP50 ↑** | **AR ↑** | **CLIP ↑** | **Aesthetic Score ↑** |
> | ------------------- | ---------- | -------- | ---------- | -------- | ---------- | --------------------- |
> | InstanceDiffusion   | 0.46       | 0.11     | 0.16       | 0.25     | 30.56      | 5.46                  |
> | IFAdapter (Direct)  | 0.69       | 0.26     | 0.47       | 0.48     | 31.53      | 5.97                  |
> | CreatiLayout        | 0.34       | 0.03     | 0.06       | 0.12     | 32.32      | 5.73                  |
> | **GAF-Pano (Ours)** | **0.70**   | 0.25     | 0.44       | **0.49** | **32.37**  | **6.12**              |
>
> - Models like InstanceDiffusion (mIoU 0.46) and CreatiLayout (mIoU 0.34) suffer from severe **Layout Collapse**.
> - While the direct IFAdapter performs relatively well, GAF-Pano still surpasses it in **Layout Fidelity** (mIoU 0.69 $\to$ 0.70). More importantly, GAF-Pano achieves a significantly higher **Aesthetic Score** (5.97 $\to$ 6.12), indicating that our joint diffusion strategy not only enforces layout but also reduces visual artifacts and enhances global coherence.
>
> We have integrated these quantitative results and the accompanying analysis into A.1.1 to provide a rigorous comparison consistent with the qualitative results initially shown in the Appendix.
>
> **4. Typographical Error (Response to Question 6)**
>
> You are absolutely correct. The term $SPM_i$ in **Equation 13** is a typographical error and should be $CPM_i$(Conditional Position Mask), as defined in **Equation 12**. We have corrected this in the revised manuscript.
>
> [1] Zhou, D., Li, Y., Ma, F., et al. "MIGC: Multi-instance generation controller for text-to-image synthesis." Proceedings of the IEEE/CVF Conference on Computer Vision and Pattern Recognition. 2024: 6818–6828.

---

### Note · Authors · 2026-03-05

I have read and agree with the venue's withdrawal policy on behalf of myself and my co-authors.

---

### Meta-Review · Area_Chair_nnEc · 2025-12-24

**Summary:**

The paper proposes GAF-Pano, a training-free framework for layout-controlled wide-aspect panorama generation using a Global Attention Fusion mechanism.

While the reviewers acknowledged the clear motivation and the effort put into the new benchmark (Pano-Layout-Bench), the consensus is that the submission falls below the bar for acceptance. The primary concerns driving this decision are the limited technical novelty relative to existing fusion methods, the restricted scope (planar vs. 360°), and mixed evidence regarding the effectiveness of the proposed Conditional Position Mask (CPM). Despite a detailed rebuttal, the reviewers remained concerned that the method represents an incremental adaptation rather than a fundamental breakthrough.

**Reviewer Concerns:**

**Addressed Concerns:**

1. Memory Overhead (4Stg): The authors provided memory profiling and inference speed analysis, clarifying that the "Sync" stage is the bottleneck but manageable.

2. Concurrent Literature (aV6y): The authors correctly clarified the timeline regarding DiT360.

3. Implementation Details (4Stg): Additional details on backbone integration (MIGC) and baseline settings were provided.

**Outstanding Concerns:**

1. Limited Novelty (ogCv): Reviewer ogCv maintained that the core contribution is somewhat limited, primarily hinging on applying the fusion mechanism from prior work (Quattrini et al. [1]) to the L2I task. The distinction from MAD was clarified but viewed as an engineering adaptation rather than a significant conceptual advance.

2. Effectiveness of CPM (ogCv): There remains significant concern regarding the Conditional Position Mask. As noted by ogCv, the quantitative results (Table 6) show that the method performs better without CPM on key metrics like mIoU. The authors' argument that standard metrics are flawed and their reliance on a small-scale manual evaluation in the rebuttal were not sufficient to strictly validate this component's design.

3. Scope and Baselines (aV6y): Reviewer aV6y highlighted the lack of comparison with modern 360° panorama generation methods (e.g., PanFusion). While the authors argued for a "planar" scope, this restriction limits the work's impact and generalizability compared to the broader state-of-the-art in panoramic synthesis.

4. Theoretical Grounding (ogCv): The admission that the "theoretical foundation" was a typo for "empirical motivation" weakened the perceived rigor of the contribution.

**Reviewer Scores:**

- Reviewer 4Stg: 4 (Maintained) or 5 (Borderline). While the memory analysis was provided, the fundamental concern about the method being an incremental "repair" to MultiDiffusion likely kept the score from crossing the acceptance threshold.
- Reviewer aV6y: 4 (Maintained). The distinction between "planar" and "spherical" panoramas, while technically valid, may have been perceived as a convenient way to avoid comparing with more advanced 360° baselines like PanFusion.
- Reviewer ogCv: 4 (Maintained). The reviewer’s core critique regarding limited novelty and the negative quantitative impact of the CPM module remained a significant hurdle that the rebuttal did not fully overcome.

---

### Decision · Program_Chairs · 2026-01-26

Reject